# Synthesizing Realistic fMRI: A Physiological Dynamics-Driven Hierarchical Diffusion Model for Efficient fMRI Acquisition

**Yufan Hu, Yu Jiang, Wuyang Li, Yixuan Yuan**[*]
The Chinese University of Hong Kong
`yufanhu@link.cuhk.edu.hk, yujiang@cuhk.edu.hk,`
`wymanbest@outlook.com, yxyuan@ee.cuhk.edu.hk`

## Abstract

Functional magnetic resonance imaging (fMRI) is essential for mapping brain activity but faces challenges like lengthy acquisition time and sensitivity to patient movement, limiting its clinical and machine learning applications. While generative models such as diffusion models can synthesize fMRI signals to alleviate these issues, they often underperform due to neglecting the brain's complex structural and dynamic properties. To address these limitations, we propose the Physiological Dynamics-Driven Hierarchical Diffusion Model, a novel framework integrating two key brain physiological properties into the diffusion process: brain hierarchical regional interactions and multifractal dynamics. To model complex interactions among brain regions, we construct hypergraphs based on the prior knowledge of brain functional parcellation reflected by resting-state functional connectivity (rsFC). This enables the aggregation of fMRI signals across multiple scales and generates hierarchical signals. Additionally, by incorporating the prediction of two key dynamics properties of fMRI—the multifractal spectrum and generalized Hurst exponent—our framework effectively guides the diffusion process, ensuring the preservation of the scale-invariant characteristics inherent in real fMRI data. Our framework employs progressive diffusion generation, with signals representing broader brain region information conditioning those that capture localized details, and unifies multiple inputs during denoising for balanced integration. Experiments demonstrate that our model generates physiologically realistic fMRI signals, potentially reducing acquisition time and enhancing data quality, benefiting clinical diagnostics and machine learning in neuroscience. Our code is available at https://github.com/CUHK-AIM-Group/PDH-Diffusion.

## 1 Introduction

Functional magnetic resonance imaging (fMRI) is a non-invasive neuroimaging technique that captures spatio-temporal patterns of blood oxygenation in the active brain (D'Esposito et al., 2003). Brain fMRI signals encapsulate the full spectrum of intrinsic functional networks and exhibit highly complex fluctuating patterns, providing very accurate information of neural activity (Strangman et al., 2002). Compared to other medical modalities, fMRI offers superior precision in predicting and diagnosing various neurological and psychiatric conditions (Matthews et al., 2006).

Despite its utility, fMRI data also presents unique challenges, particularly during acquisition. A standard fMRI scan is highly time-consuming. For example, in the data collection process of Human Connectome Project (HCP), a resting-state fMRI scan lasts 60 minutes, divided into four 15-minute sessions (Van Essen et al., 2012). While necessary for capturing comprehensive brain activity, these long sessions can be physically taxing for participants. Additionally, fMRI scans impose strict requirements on patient stillness, which is particularly difficult for infants or those unable to stay still for long periods, as even minor movements can introduce noise and artifacts, reducing data quality (Power et al., 2014; Bollmann & Barth, 2021). These challenges significantly limit its broader

---

[*]Correspondence

application in clinical diagnosis and machine learning algorithm development. *As a result, there is a growing need for methods to generate and impute fMRI time series signals, which can reduce acquisition times and optimize the quality of the acquired data.* Notably, diffusion models have recently emerged as a prominent generative approach and show promise in their generative capacity within the time series domain (Rasul et al., 2021; Shen et al., 2024; Fan et al., 2024; Li et al., 2024b;a), highlighting their potential for further exploration in fMRI signal generation.

While showing potential for application, diffusion models often underperform when applied to brain fMRI signal generation, as most advanced methods are designed for task-agnostic generation and neglect **two key intrinsic physical properties** of brain fMRI signals. First, they tend to treat fMRI data from each region-of-interest (ROI) independently, overlooking the high-dimensional interactions between signals from different ROIs (Logothetis, 2008; Bagley et al., 2017), which are indispensable for examining hypothesized disconnectivity effects in neurodegenerative and psychiatric brain diseases (Van Den Heuvel & Pol, 2010). To address this limitation, we incorporate brain regional interactions using hypergraph (see Figure 2), which is constructed based on brain rsFC[1]. Second, while these models excel at capturing temporal trends, they often fail to account for the unique physiological patterns and dynamics properties of brain signals, specifically fractal characteristics. These dynamics arise from repeated, scale-free information exchange between brain regions, generating fMRI signals that exhibit self-similarity (He, 2014). Scale-free dynamics in fMRI have been shown to vary across brain networks and behavioral conditions (Ciuciu et al., 2012), as well as change with age (Suckling et al., 2008), arousal states (Tagliazucchi et al., 2013), and disease processes (Maxim et al., 2005). Analyzing these dynamics provides critical insights into the brain mechanisms that underlie cognition and behavior (Ciuciu et al., 2014; He et al., 2023; 2024). To overcome this shortcoming, we integrate these dynamics properties into our model, emphasizing the multifractal nature of brain activity. By introducing these two fundamental aspects of brain, our method achieves more realistic and precise fMRI signal generation, better aligning with the brain's physiological reality.

The contribution of this paper can be summarized as follows: We propose a novel framework for generating brain fMRI time series signals, grounded in the above two key observations about fMRI signals, brain regional interactions, and multifractal dynamics. This framework, called the Physiological Dynamics-Driven Hierarchical Diffusion Model, is composed of three main components:

1. The Hypergraph-based Hierarchical Signals Generator: To incorporate intricate interdependencies between brain regions, we model brain rsFC as a hypergraph structure that captures complex interactions among signals of ROIs. This component aggregates fMRI signals based on the intrinsic brain functional connectivity matrix across multiple brain regions, producing hierarchical fMRI signals that encapsulate information at various scales.

2. The Dynamics Properties Guiding Module: This module is designed to incorporate the dynamics properties of brain activity, specifically utilizing the multifractal characteristics of fMRI signals into diffusion generation process. It includes a predictor that estimates the multifractal spectrum and the generalized Hurst exponent of the fMRI series. The predicted multifractal characteristics are then projected as a conditioning input to guide the diffusion process, ensuring that the generated signals maintain the complex, scale-invariant properties observed in real fMRI data.

3. The Cross-brain Region Guiding Progressive Diffusion Model: To ensure complementary signals across different brain region ranges, progressive generation is employed, where broader regional signal trends are used as conditioning inputs to guide the detailed signals of more localized brain areas. Finally, we dynamically unifies multiple conditioning inputs during the denoising phase of diffusion, ensuring balanced and coherent integration of multiple conditions.

## 2 PRELIMINARIES

### 2.1 FUNCTIONAL CONNECTIVITY OF FMRI SIGNALS.

The human brain is a complex network of functionally and structurally interconnected regions. Even at rest, there is a high level of ongoing functional connectivity and continuous information processing between the hemispheric motor cortices and between other well-established functional networks,

---

[1]brain resting-state functional connectivity (rsFC) reflects the synchronized activity and communication between brain regions during resting state.

such as the primary visual, auditory, and higher-order cognitive networks (Rogers et al., 2007; Van Den Heuvel & Pol, 2010). This leads to complex correlations between fMRI time series of ROIs on brain, represented by functional connectivity.

Functional connectivity is formally defined as the temporal dependence of neuronal activity patterns of anatomically separated brain regions (Aertsen et al., 1989; Friston et al., 1993). Functional connectivity can be described through functional connectivity matrix (Venkatesh et al., 2020), whose elements indicate the strength of functional interactions between pairs of regions, with higher values signifying stronger correlations. This matrix reflects how different regions of the brain interact or communicate and is often used to study the brain's network organization and the functional relationships underlying cognitive and physiological processes.

Given that functional connectivity reflects the complex interactions between different brain regions, multi-level partitioning provides a powerful method to analyze these interactions across varying spatial scales (Betzel & Bassett, 2017; Betzel et al., 2019). Multi-level partitioning is a common approach in brain analysis, as it allows for the simultaneous examination of brain behavior at both the macro level (e.g., interactions between brain regions) and the micro level (e.g., activity within local neuronal clusters) (Wang et al., 2021; Kan et al., 2023; Varga et al., 2024), providing a more comprehensive understanding of brain function and structure across different spatial scales.

## 2.2 MULTIFRACTALITY OF FMRI SIGNALS.

The fractal behavior has been ubiquitously observed in neuroimaging studies which may arise from various mediators such as hemodynamics, respiration, cardiac fluctuations and brain neural activities (Campbell & Weber, 2022). Extensive research has demonstrated that brain activity, regardless of the neuroimaging technique used for observation, is inherently arrhythmic and exhibits scale-free temporal dynamics (Racz et al., 2018a; Guan et al., 2022). The scale invariance dynamics in fMRI is often associated with long-range correlation in time and has been extensively demonstrated in numerous studies to be closely related to intrinsic ongoing brain activity(Ciuciu et al., 2014).

(*Definition* **Scale-free law and Self-similarity**) Data series generated by complex systems tend to fluctuate across different time scales (Boeing, 2016). These fluctuations often follow a scale-free law, maintaining consistent and invariant patterns across several orders of magnitude (Proekt et al., 2012). Scale-free dynamics can be described in the spectral domain by a power law as the power spectrum follows a single power law over all ranges of frequency.

Let $X$ denote the fMRI signals quantifying brain activity and $\Gamma_X(f)$ is its Power Spectral Density (PSD). Scale-free property is classically defined as (Ciuciu et al., 2012):

$$\Gamma_X(f) \propto f^{-\beta}, \beta \geq 0 \tag{1}$$

with $f_m \leq f \leq f_M, f_M/f_m \gg 1$, where $\beta$ is a constant parameter known as scaling exponent. The power spectrum of fMRI data follows this power law across a wide range of frequencies, suggesting that multiple frequencies equivalently contribute to its dynamics, rather than focusing solely on a specific, preselected frequency band commonly used in brain analysis. Given that the fMRI signals $X$ follow the power law as described in equation 1, we further assume that $X(t)$ is one-dimensional time series data where $t$ is the time step and $X(t)$ is a stationary jointly Gaussian process. Then the covariance function of $X(t)$ can be expressed as follows: $C_X(\tau) \sim \sigma_X^2 \left(1 + C'|\tau|^{-\alpha}\right)$, for $\tau_m \leq \tau \leq \tau_M$ with $\alpha = 1 - \beta$. $C'$ is a constant and $\sigma_X^2$ is variance of $X$. Then it is easily to derive that: $\mathbb{E}(X(t+\tau) - X(t))^2 = \mathbb{E}X(t+\tau)^2 + \mathbb{E}X(t)^2 - 2\mathbb{E}X(t+\tau)X(t) = c_2|\tau|^{-\alpha}$, where $c_2 = -2\sigma_X^2 C'$. The fact that $X$ is Gaussian further suggests that $\forall q > -1$:

$$\mathbb{E}|X(t+\tau) - X(t)|^q = c_q|\tau|^{-\frac{q\beta}{2}}, \tau_m \leq \tau \leq \tau_M \tag{2}$$

Defining $Y(t) = \int^t X(s)ds$, equation is as follows, when $\tau_m \leq \tau_1, \tau_2 \leq \tau_M$:

$$\left\{ \frac{Y(t+\tau_1) - Y(t)}{\tau_1^H} \right\}_{t \in \mathbb{R}} \stackrel{\text{fdd}}{=} \left\{ \frac{Y(t+\tau_2) - Y(t)}{\tau_2^H} \right\}_{t \in \mathbb{R}} \tag{3}$$

Where $H = (-\alpha/2) = (\beta-1)/2$ and $\stackrel{\text{fdd}}{=}$ means equality of all joint finite dimensional distributions. In other words, this means that for all $q > -1$, such that $\mathbb{E}|Y(t)|^q < \infty$:

$$\mathbb{E}|Y(t+\tau) - Y(t)|^q = c_q|\tau|^{qH}, \tau_m \leq \tau \leq \tau_M, \text{ or}$$
$$\mathbb{E}|Y(t+\tau_2) - Y(t)|^q = \mathbb{E}|Y(t+\tau_1) - Y(t)|^q \left(\frac{|\tau_2|}{|\tau_1|}\right)^{qH}, \tag{4}$$

when $\tau_m \leq \tau_1, \tau_2 \leq \tau_M$.

A geometric dataset exhibiting scale invariance is considered self-similar if it can be decomposed into smaller parts, each of which resembles the entire original structure (MIshra & Bhatnagar, 2014). As shown in equation 3 and equation 4, $Y(t)$ is an example of a self-similar process.

From a more generalized perspective, above equations are not only fold for jointly Gaussian process but for a broader and more general class, that of self-similar processes with stationary increments, referred to as H-sssi processes, and defined in (Samorodnitsky et al., 1996):

$$\{X(t)\}_{t\in\mathbb{R}} \stackrel{\text{fdd}}{=} \{a^H X(t/a)\}_{t\in\mathbb{R}} \tag{5}$$

For $\forall a > 0$, $H \in (0, 1)$. Parameter $H$ is referred to as the self-similarity exponent.

For real physiological data collected from brain, fMRI data can be also be viewed as the increment process $Y(t) = X(t + \tau_0) - X(t)$ of an H-sssi process $X$, where $\tau_0$ is a constant chosen by physiology and data acquisition set up, thereby exhibiting both scale-free and self-similarity properties.

Data from H-sssi processes can be typically characterized by monofractal scaling exponents. However, in reality, brain signals are more complex: while they exhibit global pattern consistency (monofractality), they also demonstrate distinct functional activity patterns in local areas (multifractality) (Racz et al., 2018b; França et al., 2018; La Rocca et al., 2018). This reflects the multi-scale regulation of neural activity in the brain. Larger-scale signals may capture whole-brain functional coordination, aligning with the global self-similarity described by monofractal models, whereas smaller-scale signals represent localized neuronal group activity.

(*Definition* **Multifractality**) For fMRI data $Y(t)$, equation 1 holds over a wide range of $\tau$. However, the scaling exponent deviate significantly from the expected linear behavior $qH$, manifesting as:

$$\mathbb{E}|Y(t + \tau) - Y(t)|^q = c_q |\tau|^{\zeta(q)}, \tau_m \leq \tau \leq \tau_M \tag{6}$$

Note the $q$-order scaling exponent $\zeta(q)$ is necessarily a strictly concave function of $q$. For this reason, instead of H-sssi process, there need a broader class to depict this process, referred to as that of multifractal processes.

Data with multifractal properties are more complex, featuring varying local characteristics and described by a range of scaling exponents. Multifractal signals exhibit both small-scale and large-scale local fluctuations, which are absent in monofractal signals. These fluctuations, associated with different statistical moments, enable multifractals to capture fractal properties across multiple scales and represent localized nonlinear dynamics within the data (Lopes & Betrouni, 2009). By extending this capability, multifractal models provide a deeper understanding of brain dynamics, characterizing the interactions across various scales and revealing how cognitive functions emerge from the synergy of processes operating at multiple levels.

## 3 METHOD

### 3.1 THE FRAMEWORK

The overall framework of the Physiological Dynamics-Driven Hierarchical Diffusion Model is illustrated in Figure 1. The fMRI time series data of a subject with $N$ ROIs can be denoted as $\mathbf{X}_T = (x_1, \ldots, x_i, \ldots, x_N)^T \in \mathbb{R}^{N \times T}$, where $x_i \in \mathbb{R}^T$ represents the time series of the $i$-th ROI. Each $x_i$ spans $T$ timesteps in the time series. We are given input data $X_{t_0-L:t_0} \in \mathbb{R}^{N \times L}$, where $L$ represents the size of the retrospective window, and $t_0$ denotes the initial position of the forecast window. The objective of the task is to predict the future fMRI values of $N$ ROIs for a time span of $t$ future time steps $X_{t_0:t_0+t} \in \mathbb{R}^{N \times t}$.

First, we input the fMRI time series data $X_{t_0-L:t_0}$ into the Hypergraph-based Hierarchical Signal Generator to produce $R$ time series data $X_{t_0-L:t_0}^r$, where $r$ denotes signals aggregated from $r$-level spatial ranges of brain regions. Then, we use a diffusion model to generate $R$ time series data $\hat{X}_{t_0:t_0+t}^r$ signals. Specifically, we first extract the historical embedding $\mathbf{h}_{t_0}^r = \text{RNN}_\theta \left(\boldsymbol{x}_{t_0}^r, \mathbf{h}_{t_0-1}^r\right)$ of the known time window using RNN, which serves as the basic historical condition $c_{history}$ for the diffusion process. This embedding is then input into the Dynamics Properties Guiding Module,

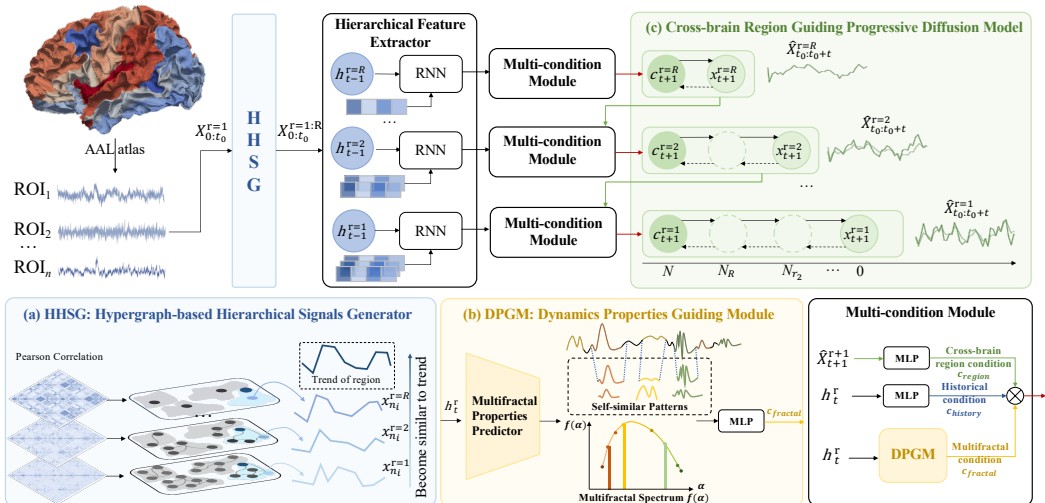

Figure 1: The framework of Physiological Dynamics-Driven Hierarchical Diffusion Model (PDH-Diffusion) with three main modules (a, b, and c), where we introduce two key physiological characteristics of brain fMRI to generate more realistic fMRI signals.

where a specifically designed loss function $L_{fractal}$ is used to optimize the predicted multifractal characteristics and generate the corresponding multifractal conditions $c_{fractal}$. Additionally, in the Cross-brain Region Guiding Progressive Diffusion Model, signals from broader brain region ranges $\hat{X}_{t_0:t_0+t}^{r+1}$ are used as cross-brain region conditions $c_{region}$ to guide the generation of more detailed signals $\hat{X}_{t_0:t_0+t}^{r}$ during the diffusion process. As a result, we obtain realistic generated signals $\hat{X}_{t_0:t_0+t}^{r=1}$, which accurately capture brain region relationships, fractal characteristics, while maintaining alignment with the known retrospective window data.

## 3.2 HYPERGRAPH-BASED HIERARCHICAL SIGNALS GENERATOR

The first key module in our framework is the Hypergraph-based Hierarchical Signals Generator, where we model the complex interdependence between fMRI signals from sample points across different brain regions as a hypergraph structure. This structure enables the propagation of information across varying spatial scales of brain regions, resulting in hierarchical fMRI signals that capture multiple levels of information.

As mentioned before, the fMRI time series data of a subject with $N$ ROIs is denoted as $\mathbf{X} = (x_1, \ldots, x_i, \ldots, x_N)^T \in \mathbb{R}^{N \times T}$. Traditional generation methods (Rasul et al., 2021; Alcaraz & Strodthoff, 2022) fail to capture the complex high-dimensional physiological and structural dependencies between fMRI data of ROIs, leading to suboptimal outcomes. In this framework, we address this by modeling the relationships between fMRI signals from different ROIs using a hypergraph structure.

To align with real physiological conditions, we first model the fMRI signals as a standard graph structure $\mathcal{G}_s$, based on the functional connectivity matrix $C \in \mathcal{R}^{N \times N}$ as the adjacency matrix, where $N$ is the number of ROIs and each element in $C$ reflects the interaction patterns between different ROIs (i.e., between distinct fMRI signals). In particular, a threshold is applied to the value distribution in $C$, where edges are established between sampling points that exhibit connectivity values exceeding the threshold. The resulting graph is denoted as $\mathcal{G}_s = (\mathcal{V}_s, \mathcal{E}_s)$, where $v_i \in \mathcal{V}_s$ denotes a vertex corresponding to a sampling point, and $e_{ij} \in \mathcal{E}_s$ represents an edge connecting vertices $v_i$ and $v_j$.

On the basis of constructed graph $\mathcal{G}_s$ of fMRI signals, we further construct a hypergraph $\mathcal{G} = (\mathcal{V}, \mathcal{E}, W)$ by defining hyperedges using the $k$-Hop neighbors method as described in (Gao et al., 2022). The hypergraph $\mathcal{G}$ consisting of a vertex set $\mathcal{V}$, a hyperedge set $\mathcal{E}$, and a hyperedge weight

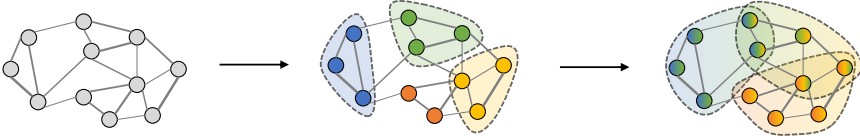

standard graph $\mathcal{G}_s$     hypergraph $\mathcal{G}^r$ with neighbor $k_r$     hypergraph $\mathcal{G}^{r+1}$ with neighbor $k_{r+1}$

Figure 2: The construction of hierarchical hypergraph. Starting from a standard graph $\mathcal{G}_s$ constructed using rsFC as the adjacency matrix, hypergraphs with different aggregation levels $\mathcal{G}^r$ are built using a $k$-hop method. As the number of neighbors $k_r$ increases, each node (fMRI signal of a certain ROI) integrates more intra-regional and, gradually, inter-regional information (with different brain regions' information represented by different colors).

matrix $W$ and can be represented by its incidence matrix $H \in \{0,1\}^{|\mathcal{V}| \times |\mathcal{E}|}$, where the entries of $H$ are defined as: $H(v,e) = \begin{cases} 1, & \text{if } v \in e, \\ 0, & \text{otherwise.} \end{cases}$. We form hyperedge group using $k$-Hop neighbors $\mathcal{E}$ which seeks to identify vertices related to a central node through $k$-Hop reachability within the graph and group them as a hypervertex. The $k$-Hop neighborhoods of a vertex $v$ in graph $\mathcal{G}_s$ is defined as: $N_{\text{hop}_k}(v) = \{u \mid \mathbf{C}_{uv}^k \neq 0, u \in \mathcal{V}_s\}$. Here $k$ can vary from $[2, n_v]$, where $n_v$ is the number of vertices in $\mathcal{G}_s$. Finally, the hyperedge group with $k$-Hop can be written as: $\mathcal{E}_{\text{hop}_k} = \{N_{\text{hop}_k}(v) \mid v \in \mathcal{V}\}$. Additionally, existing brain parcellation techniques can also be used to construct a hypergraph Li et al. (2023), which may vary depending on the task performed during fMRI acquisition. In the experiment, we select a neighbor list $K = \{k_1, ..., k_R\}$, where each neighbor $k_r$ is used to construct a corresponding hypergraph $\mathcal{G}^r$. We construct $R$ hypergraphs, each representing information from a distinct range of brain areas, as shown in Figure 2.

In hypergraph propagation process, a signal takes the aggregation of messages from its inter neighbor set vertices to get a new representation of the central vertex. We leverage the two steps of hypergraph message propagation method in (Gao et al., 2022), which is defined as two stage:

The first stage is: $\begin{cases} m_\beta = \sum\limits_{\alpha \in \mathcal{N}_v(\beta)} M_v(x_\alpha) \\ y_\beta = U_e(w_\beta, m_\beta) \end{cases}$, and the second stage is: $\begin{cases} m'_\alpha = \sum\limits_{\beta \in \mathcal{N}_e(\alpha)} M_e(x_\alpha, y_\beta) \\ x'_\alpha = U_v(x_\alpha, m'_\alpha) \end{cases}$.

where $x_\alpha \in \mathbf{X}$ is the signal of vertex $\alpha \in \mathcal{V}$, and $x'_\alpha$ is the updated signal of vertex $\alpha \in \mathcal{V}$ after information propagation. $m_\beta$ is the message of hyperedge $\beta \in \mathcal{E}$ and $m'_\alpha$ is the message of vertex $\alpha$. $y_\beta$ is the representation of hyperedge $\beta$. $M_v(\cdot), U_e(\cdot), M_e(\cdot), U_v(\cdot)$ are the vertex message function, hyperedge update functions, hyperedge message function and vertex update function respectively. In experiment, we choose the basic implement as message function and update functions, both of which are mean functions and the propagation step is set to 1.

By constructing fMRI data as hypergraphs at various scales, we can obtain multiple sets of fMRI data $\mathbf{X}^R \in \mathcal{R}^{R \times N \times T}$. FMRI data aggregated from larger regions capture broader brain area information, while data from smaller regions provide more detailed, localized insights within the region. To fully utilize these hierarchical fMRI signals and ensure complementary information across different brain regions, we design a cross-brain region guiding module, as detailed in Section 3.4.

### 3.3 DYNAMICS PROPERTIES GUIDING MODULE

To ensure that the fMRI signals generated by the diffusion model exhibit the same dynamic characteristics as real brain physiological signals, we design a module in which a multifractal properties predictor is optimized to estimate the multifractal characteristics of the signal over a future time window. The multifractal condition is then obtained through a projection layer and serves as one of the multiple conditions guiding the signal generation in the diffusion model.

We perform multifractal analysis on the fMRI signals and extract their fractal characteristics to serve as ground truth. Multifractal detrended fluctuation analysis (MFDFA) is a widely used method for analyzing the multifractal properties of time series, with detailed studies available in (Telesca et al., 2016; Ihlen, 2012). This method provides various indicators to characterize the fractal nature of the

time series. From these, we selected two representative and commonly used characteristics. Their descriptions are provided below:

1. The generalized Hurst exponent $H(q)$: It reflects the inhomogeneity of the signal's structure. For monofractals, $H(q)$ is independent of the parameter $q$ and remains constant, whereas for multifractals, $H(q)$ varies with changes in $q$.

According to definition in Equation 6, $H(q)$ can be derived from the $q$-order scaling exponent $\zeta(q)$:

$$\zeta(q) = qH(q) - D_T \tag{7}$$

where $D_T$ is the topological dimension, which equals 1 for time series.

2. The multifractal spectrum $f(\alpha)$: It implies the variation of the local Hurst exponent. The generalized fractal dimension $D(q)$ and multifractal spectrum $f(\alpha)$ can be derived as:

$$D(q) \equiv \frac{\zeta(q)}{q-1}, \quad \alpha = \zeta'(q), \quad f(\alpha) = q\alpha - \zeta(q) \tag{8}$$

Note that $\alpha$ and $f(\alpha)$ are also referred to as $q$-order singularity exponent $hq$ and $q$-order singularity dimension $Dq$ in (Ihlen, 2012), as $h$ and $D(h)$ in other literature (Ihlen & Vereijken, 2010).

Though the multifractal properties of fMRI time series can be calculated by MFDFA during the training phase, in practical inference, no direct observations are available for the signals in future window. To address this, we design a multifractal properties predictor to forecast the multifractal characteristics of future signals. The prediction is based on the historical embedding $\mathbf{h}_t^r = \text{RNN}_\theta\left(\boldsymbol{x}_t^r, \mathbf{h}_{t-1}^r\right)$ which is extracted by RNN model from the observed window of the time series.

During training, we optimize the predicted generalized Hurst exponent $\hat{H}(q)$ using the Mean Squared Error (MSE) loss, a commonly used loss function for regression tasks. Additionally, to ensure the predicted multifractal spectrum $\hat{f}(\alpha)$ closely approximates the true distribution, we employ the Kullback-Leibler (KL) divergence $D_{KL}(\hat{f}(\alpha)||f(\alpha))$ to measure the difference between the predicted spectrum $\hat{f}(\alpha)$ and the ground truth spectrum $f(\alpha)$, which is calculated using MFDFA.

The complete fractal loss for a certain fMRI data $x^r$ is:

$$L_{fractal}^{(r)}(\gamma) = \frac{1}{n}\sum_i^n (H_i(q) - \hat{H}_i(q))^2 + \sum_i^m f_i(\alpha)\log\frac{f_i(\alpha)}{\hat{f}_i(\alpha)} \tag{9}$$

where $n$ and $m$ is the dimension of calculated generalized Hurst exponent and multifractal spectrum.

After obtaining the predicted multifractal properties, we project them using a multifractal projector, which can be implemented as a MLP, and obtain the multifractal condition $c_{fractal}$ that captures the multifractal dynamics of fMRI data.

## 3.4 Cross-brain region guiding progressive diffusion model

We aggregate signals from brain regions of different ranges to complement the information in diffusion generation process, as signals from broader brain regions capture overall brain activity trends, while those from smaller regions provide finer details within a region. According to previous studies on brain function (Weiskopf et al., 2004; Kriegeskorte & Bandettini, 2007), different brain regions respond to various tasks with distinct activation states, reflected in the overall increase or decrease trend of fMRI signals. In diffusion model, the broader trend signals correspond to the early generation steps, representing brain region activity at a larger scale. Therefore, we generate fMRI signals that represent the general activity trends of broader brain regions, and then use these trends as conditions to guide the generation of finer-scale signals. This multi-level, progressive approach ensures that signals at each level collectively contribute to refining the generation of the final fMRI signal.

Intuitively, the signal representing the trend across a broader range of brain regions tends to be smoother, as it integrates multiple fMRI signals that exhibit similar trends within that region. As a result, fewer diffusion generation steps are required. We also set up share ratio as in (Fan et al., 2024), defined as: $\sigma_r := 1 - (N_*^r - 1)/N$, which represents the shared percentage of variance schedule between the data aggregated from $r$-level brain region and the original data. For the original fMRI

data, $N_*^1 = 1$ and $\sigma_1 = 1$.

$$\alpha_n^r (N_*^r) = \begin{cases} 1 & \text{if } n = 1, \ldots, N_*^r \\ \alpha_n^1 & \text{if } n = N_*^r + 1, \ldots, N \end{cases} \tag{10}$$

Here, $\{\beta_n^r\}_{n=1}^N = \{1 - \alpha_n^r\}_{n=1}^N$ is the variance schedule in diffusion and $a_n^r (N_*^r) = \prod_{k=1}^n \alpha_k^r$, $b_n^r (N_*^r) = 1 - a_n^r (N_*^r)$. Noted that aligning with the intuition, there is $N_*^1 < N_*^2 \ldots < N_*^r < \ldots < N_*^R$ for brain region level $r \in \{1, ..., R\}$.

Unlike the conventional multi-granularity diffusion generation process, in the case of brain fMRI generation, signals representing different spatial scales of brain regions play a clearer guiding role. We first generate signals that represent the overall active trend of brain regions, which then guide the generation of finer detailed signals within each region. To achieve this, we establish a progressive generation framework. Specifically, when generating the signal at the $r$-th level, we leverage $(r+1)$-th level generated signal through extracting its embedding and then mapping it by a MLP, as a cross-brain region condition $c_{region}$ to guide the diffusion generation.

### 3.5 TRAINING PROCEDURE

For the signal aggregated from $r$-range of brain regions, we train the conditional denoising diffusion model with condition $\mathbf{c}^r = w_h c_{history}^r + w_f c_{fractal}^r + w_r c_{region}^r$. We have diffusion loss $L^{(r)}(\theta)$ at timestep $t$ and diffusion step $n$ as below, where $\boldsymbol{\epsilon} \sim \mathcal{N}(\mathbf{0}, \boldsymbol{I})$:

$$L_{diffusion}^{(r)}(\theta) = \mathbb{E}_{\boldsymbol{\epsilon}, \boldsymbol{x}_{0,t}^r, n} \| (\boldsymbol{\epsilon} - \boldsymbol{\epsilon}_\theta (\sqrt{a_n^r} \boldsymbol{x}_{0,t}^r + \sqrt{b_n^r} \boldsymbol{\epsilon}, n, \mathbf{c}^r) \|_2^2, \tag{11}$$

The final training objective is a weighted sum of diffusion loss in Equation 11 and the fractal loss in Equation 9 for all signals:

$$
\begin{aligned}
L^{\text{final}} &= \sum_{r=1}^R \omega^r L_{diffusion}^{(r)}(\theta) + \sum_{r=1}^R L_{\text{fractal}}^{(r)}(\gamma) \\
&= \sum_{r=1}^R \omega^r \mathbb{E}_{\boldsymbol{\epsilon}, \boldsymbol{x}_{0,t}^r, n} [\| \boldsymbol{\epsilon} - \boldsymbol{\epsilon}_\theta (\boldsymbol{x}_{n,t}^r, n, \mathbf{c}^r) \|_2^2] + \sum_{r=1}^R L_{\text{fractal}}^{(r)}(\gamma),
\end{aligned}
\tag{12}
$$

where $\boldsymbol{x}_{n,t}^r = \sqrt{a_n^r} \boldsymbol{x}_{0,t}^r + \sqrt{b_n^r} \boldsymbol{\epsilon}$ and $\sum_{r=1}^R \omega^r = 1$ is hyper-parameter controlling the scale of guidance as in (Fan et al., 2024).

## 4 EXPERIMENT

### 4.1 SETTINGS

**Datasets**

To evaluate the advanced capabilities of proposed fMRI generation method, we choose the Human Connectome Project (HCP) dataset 900 Subject Release (Van Essen et al., 2012; 2013), which is a public fMRI dataset for brain related research. We preprocess the fMRI data using standard methods (Glasser et al., 2013) and parcellate it into nodes using a whole-brain functional AAL atlas, from which we select 86 nodes in the cerebrum. The brain functional connectivity matrix is constructed by calculating the Pearson correlation between the fMRI time series of different brain ROIs.

**Baselines and Evaluation Metrics**

As our model is diffusion-based time series generation model, we mainly choose two types of baselines to compare. (1) we choose time series forecasting methods, such as TimesNet (Wu et al., 2023), Flowformer (Huang et al., 2022), iTransformer (Liu et al., 2023), MSGNet (Cai et al., 2024), U-Mixer (Ma et al., 2024); (2) we include recent time series diffusion models: non-autoregressive diffusion model TimeGrad (Rasul et al., 2021), structured state space model-based diffusion SSSD (Alcaraz & Strodthoff, 2022).

We evaluate our model and all baselines using MAE (Mean Absolute Error), MAPE (Mean Absolute Percentage Error) and RMSE (Root Mean Square Error), which are widely used metrics for accuracy of time series forecasting.

Figure 3: Prediction performance of fMRI signals across 32 and 96 time steps. Each row represents a different subject, and each column corresponds to a distinct ROI.

Table 1: Comparison on fMRI forecasting task on HCP dataset. Baseline methods include time series forcasting methods (TimesNet (Wu et al., 2023), Flowformer (Huang et al., 2022), iTransformer (Liu et al., 2023), MSGNet (Cai et al., 2024), U-Mixer (Ma et al., 2024)) and the diffusion-based method (TimeGrad (Rasul et al., 2021), SSSD (Alcaraz & Strodthoff, 2022)). ↓ indicates smaller values are preferred. Bold font indicates the best result in a column.

| Method | $T_{pred}=32$ | | | $T_{pred}=64$ | | | $T_{pred}=96$ | | |
| | MAE ↓ | RMSE ↓ | MAPE(%) ↓ | MAE ↓ | RMSE ↓ | MAPE(%) ↓ | MAE ↓ | RMSE ↓ | MAPE(%) ↓ |
|---|---|---|---|---|---|---|---|---|---|
| TimesNet (Wu et al., 2023) | 31.17 | 42.12 | 0.2956 | 31.39 | 42.64 | 0.2998 | 31.70 | 42.82 | 0.3015 |
| Flowformer (Huang et al., 2022) | 30.55 | 40.89 | 0.2917 | 30.74 | 40.16 | 0.2953 | 30.97 | 40.28 | 0.2974 |
| iTransformer (Liu et al., 2023) | 32.64 | 43.54 | 0.3166 | 32.52 | 43.59 | 0.3183 | 32.77 | 43.82 | 0.3214 |
| MSGNet (Cai et al., 2024) | 31.21 | 42.53 | 0.2981 | 31.56 | 42.82 | 0.3015 | 31.95 | 42.90 | 0.3022 |
| U-Mixer (Ma et al., 2024) | 29.36 | 39.70 | 0.2793 | 29.42 | 39.81 | 0.2804 | 29.83 | 39.92 | 0.2817 |
| TimeGrad (Rasul et al., 2021) | 34.47 | 47.79 | 0.3413 | 36.42 | 49.13 | 0.3527 | 37.62 | 50.35 | 0.3720 |
| SSSD (Alcaraz & Strodthoff, 2022) | 34.38 | 47.06 | 0.3394 | 35.18 | 48.21 | 0.3406 | 36.27 | 49.31 | 0.3685 |
| PDH-Diffusion (Ours) | **28.56** | **38.70** | **0.2744** | **29.04** | **38.91** | **0.2768** | **29.73** | **39.25** | **0.2803** |

## Experimental Setups

There are 696 data in training dataset and 174 data in test dataset. Each data has size $N_{ROI} \times T$, where $N_{ROI} = 82$ is the number of ROIs on brain, $T = 1200$ is the timesteps of fMRI time series. For fMRI time series prediction, we set the context length $L = 64$, and prediction length $T_{pred} = \{32, 64, 96\}$. These settings were applied to all compared models. The initial learning rate was $LR = 0.00001$ and batch size was 32. The training was conducted for 200 epochs.

## 4.2 RESULTS AND ANALYSIS

The MAE, RMSE, and MAPE values, averaged over 10 independent runs, are reported in Table 1. From the results, we can see for all three prediction time length $T_{pred} = 32$, $T_{pred} = 64$ and $T_{pred} = 96$, PDH-Diffusion consistently outperforms the baselines in terms of MAE, RMSE, and MAPE. We also observe that as the prediction length increases while keeping the context length constant, the forcasting accuracy of all models declines to varying degrees. We show the prediciton performance of fMRI data in Figure 3. Overall, the result indicates that our method can generate fMRI that are closest to the true physiological fMRI signals.

## 4.3 ABLATION ANALYSIS

To verify the rational business of our framework, we provide detailed ablations. In particularly, we conducted three separate ablation experiments, each corresponding to the removal of one of the proposed modules: (1)w/o-HHSG, i.e. Model without the Hypergraph-based Hierarchical Signals Generator from the framework, meaning that the diffusion model was only trained to generate the original fMRI data without any cross-brain region information. (2)w/o-DPGM, i.e. Model without the Dynamics Properties Guiding Module, which also eliminates the need for the loss function $L_{fractal}$ during training. (3)w/o-CRGM, i.e. Model with the Cross-brain Region Guiding Progressive Diffusion Model removed, which means there is no cross-brain region condition derived from the wider range of brain regions. We evaluated the whole framework with these three variants on HCP dataset and the results are reported in Table 2, highlighting improvements of proposed modules.

**The influence of the weights of condition $c_{fractal}$ and $c_{region}$**

In the proposed framework, the diffusion generation process is guided by multiple conditions, with the historical condition as the base and the cross-brain region and multifractal conditions added.

Table 2: Ablation study covering removing three key components proposed in framework.

| Metric | MAE ↓ | RMSE ↓ | MAPE (%) ↓ |
|---|---|---|---|
| **PDH-Diffusion(Full)** | **29.04** | **38.91** | **0.2768** |
| **w/o-HHSG** | 34.26 | 44.93 | 0.3397 |
| **w/o-DPGM** | 33.72 | 43.55 | 0.3356 |
| **w/o-CRGM** | 31.08 | 42.47 | 0.3218 |

Table 3: The influence of the brain region range setting in HHSG.

| Number of ranges | MAE ↓ | RMSE ↓ | MAPE(%) ↓ |
|---|---|---|---|
| **1** | 34.26 | 44.93 | 0.3397 |
| **2** | 33.89 | 44.25 | 0.3314 |
| **3** | 31.08 | 42.53 | 0.3132 |
| **4** | 29.86 | 40.24 | 0.2893 |
| **5** | **29.04** | **38.91** | **0.2768** |

To evaluate the importance of these additional conditions, we explored the impact of their weights $w_r$ and $w_f$ on the generation results, as presented in Figure 4a.

As shown in Figure 4a, for both conditions, the results initially decrease and then increase as the weight grows. The initial decline reflects the effectiveness of the added conditions, while the subsequent increase may be due to excessive weighting interfering with the basic (history) condition. Additionally, it demonstrates that the fractal condition consistently improves the generation results.

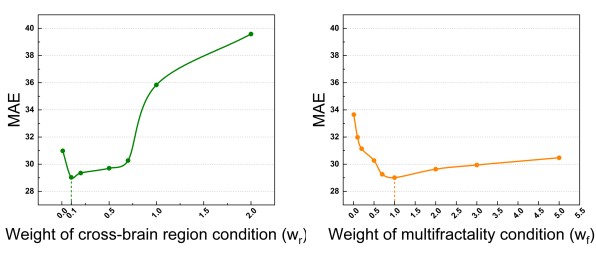 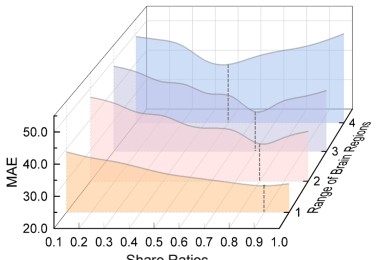

(a) The impact of the weights of cross-brain region condition and multifractal condition.

(b) The influence of share ratios on the generation of signals from different brain ranges.

Figure 4: The impact of weights of conditions and share ratios.

### The influence of share ratios $\sigma$

To assess the impact of the share ratios $\sigma$ on signal generation across different brain region ranges, we conducted an experiment with four ranges ($R = 4$), as shown in Figure 4b. The share ratio for signal from each range was varied from 0.1 to 1. In the figure, $r = 1$ represents the original, most localized signal, while $r = 4$ corresponds to the signal from the widest brain region range.

Figure 4b shows that for the original signal, which contains more detailed local information, a higher share ratio leads to better generation performance. In contrast, for signals that aggregate more global information and represent broader brain region trends, the optimal share ratio tends to be lower.

### The influence of the brain region range setting

In Table 3, we experimentally examined the effect of varying the number of ranges of brain regions on the generated results. As shown in Table 3, the generation performance improves as the number of ranges of brain regions increases, indicating that richer and more detailed transmission of brain region information significantly enhances the quality of generated fMRI signals. This finding further suggests that incorporating additional medical-based prior knowledge about brain partitions holds great potential for further improving fMRI signal generation.

## 5 CONCLUSION

In this paper, we propose the Physiological Dynamics-Driven Hierarchical Brain Diffusion Model, a novel framework for generating realistic fMRI time series by integrating key aspects of brain functional connectivity and multifractal dynamics. By leveraging a hypergraph-based functional connectivity structure and a multifractal guiding module, our approach improves the physiological accuracy of generated fMRI signals, addressing limitations in current generative models and offering potential for enhanced applications in neuroscience and clinical diagnosis.

## 6 ACKNOWLEDGMENTS

This work was supported by Hong Kong Research Grants Council (RGC) General Research Fund 14220622 and CUHK 4055188.

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
