# OpenReview forum: "Synthesizing Realistic fMRI: A Physiological Dynamics-Driven Hierarchical Diffusion Model for Efficient fMRI Acquisition"
_ICLR.cc/2025/Conference — ICLR 2025 Poster_

### Official Review · Reviewer_ouRH · 2024-10-31

**Soundness:** 3
**Presentation:** 3
**Contribution:** 3
**Rating:** 8
**Confidence:** 4

**Summary:**

In this study, the authors proposed a novel algorithm for synthesizing realistic functional MRI (fMRI) via physiological dynamics-driven hierarchical diffusion model. Then, the authors validated the feasibility of synthesized fMRI data by comparing it to other generative mechanisms. The scientific merit of this work mainly comes from the conceptual advances in their proposed algorithm. It is indeed challenging to synthesize realistic fMRI data while preserving unique aspects of the brain system. The authors combined three different modules, each serving different roles, to synthesize brain dynamics with preserved network-like structure and fractal components. Given results of the extensive validation experiment on the large fMRI cohort, the validity of the proposed algorithm is clear; yet, the scientific significance of this work is somewhat limited as there was no experiment demonstrating the practical usefulness of the proposed algorithm.

**Strengths:**

The major originality of this study comes from its conceptual advances embedded in the proposed algorithm. It is indeed challenging to synthesize realistic fMRI data while preserving unique aspects of the brain system. The quality and clarity of the models in the main text are reasonably strong as well.

**Weaknesses:**

The major concern comes from the unclear significance of this work. As the authors argued in the Introduction, acquisition of fMRI is expensive. Thus, synthesizing fMRI signal can be tempting. Although expensive, real fMRI data reflects unique information of individuals. This study, however, was not able to demonstrate the synthesized fMRI still convey unique information of subjects. Slight improvement in forecasting future timepoints of fMRI signal does not suggest the significance or practical usefulness of the model. This works needs additional analysis or applications highlighting the unique advantage of the synthesized fMRI data from the proposed model.

**Questions:**

1. In table 1, does “T_Pred=32 or 64” mean predicting 32 or 64 time points of fMRI data?
2. If my speculation in the former question is right, it is counter-intuitive that forecasting more time points (96 vs. 32) did not lead to an increase in errors. Please clarify this.
3. According to Fig 1, it looks like the resolution of synthesized data is bounded to the level of ROI. It is more desirable to synthesize fMRI dynamics at the level of vertex. Is the proposed method able to be applied at the vertex level as well?
4. In line 451, please double check the definition of MAE.

---

> ### Author Response · Authors · 2024-11-20
> **Response to Reviewer ouRH (Part 1)**
>
> We sincerely appreciate your thorough review and thoughtful suggestions. Your feedback is incredibly valuable and has greatly contributed to enhancing the quality of our work. Below, we have provided detailed responses to each of your questions and concerns, and we hope these explanations address your queries effectively.
>
> > Question1: In table 1, does “T_Pred=32 or 64” mean predicting 32 or 64 time points of fMRI data?
> If my speculation in the former question is right, it is counter-intuitive that forecasting more time points (96 vs. 32) did not lead to an increase in errors. Please clarify this.
>
> Your understanding is certainly correct. We may not have explained it clear enough in the paper. The three evaluation metrics (MAE,RMSE,MAPE) we used are such that smaller values indicate better performance. As shown in Table 1 in the paper, with an increase in the prediction length, the prediction performance of all models slightly decreases, which aligns with intuitive expectations. Thank you for pointing this out. If we have not addressed your concerns clearly, please do not hesitate to let us know. We greatly appreciate your suggestions and are happy to provide further clarification.
>
> > Question2: According to Fig 1, it looks like the resolution of synthesized data is bounded to the level of ROI. It is more desirable to synthesize fMRI dynamics at the level of vertex. Is the proposed method able to be applied at the vertex level as well?
>
> Thank you for your thoughtful question. Our method is adaptable to other brain parcellation atlases, and higher resolution can be achieved by utilizing atlases with a greater number of ROIs. We fully agree that synthesizing fMRI dynamics at the vertex level would be more desirable. However, there are significant challenges in doing so, primarily stemming from limitations in the data itself. Below, we outline some key reasons for focusing on ROI-level synthesis in our current work:
>
> 1. Massive Data Scale at Vertex-Level fMRI and Computational Challenges: As highlighted in the HCP data processing pipeline[1], the raw data at the voxel or grayordinates level results in massive correlation matrices, with sizes of about 190 GB for voxels and 32 GB for grayordinates. Such large-scale data pose significant computational challenges. Our method focuses on ROI-level synthesis to balance meaningful brain dynamics and computational feasibility. Extending to vertex-level synthesis is theoretically possible but would require addressing significant computational challenges.
>
> 2. Biological Interpretability and Functional Relevance: ROI-level fMRI signals align well with known biological functions, leveraging the brain's hierarchical organization as defined by established atlases. This abstraction enables us to incorporate prior knowledge of brain regions and their functional roles, making it easier to interpret results and apply them to downstream analyses such as functional connectivity or network modeling. Vertex-level synthesis would require extensive effort to relate raw voxel signals to meaningful biological processes.
>
> 3. Cross-Subject Alignment: Individual variations in brain anatomy and function present a significant challenge at the voxel level. ROI-based fMRI, defined by a brain atlas, aligns data to a common coordinate space, allowing for consistent cross-subject comparisons and reducing the impact of individual anatomical differences. Without this alignment, synthesizing vertex-level dynamics would suffer from greater variability across subjects, complicating both analysis and interpretation.
>
> While these reasons guide our focus on ROI-level fMRI synthesis in the current work, we acknowledge the potential value of vertex-level fMRI synthesis. Our method could theoretically be extended to this finer scale, but doing so would require addressing these challenges, such as designing methods to incorporate brain region prior knowledge and developing methods to improve cross-subject alignment at the voxel level.
>
> [1] Van Essen, David C., et al. "The WU-Minn human connectome project: an overview." Neuroimage 80 (2013): 62-79.

---

> ### Author Response · Authors · 2024-11-20
> **Response to Reviewer ouRH (Part 2)**
>
> > Question3: In line 451, please double check the definition of MAE.
>
> Thank you for pointing out this issue. Initially, we intended "Moving Absolute Error" as a metric to capture the rolling or dynamic absolute error over time, considering that fMRI signals are inherently dynamic. Moving Absolute Error captures these temporal variations by focusing on smaller, localized segments, providing a finer-grained error analysis compared to Mean Absolute Error, which aggregates errors over the entire time series and might mask transient deviations. However, we acknowledge that in our current implementation, the window size is equal to the prediction length, meaning the "moving" aspect was not explicitly reflected in the metric. As a result, the definition we used aligns exactly with the standard "Mean Absolute Error (MAE)" commonly used in other time-series prediction studies. Similarly, the definition we used for Moving Absolute Percentage Error (MAPE) in our experiments is equivalent to the standard Mean Absolute Percentage Error (MAPE) commonly employed in other time-series prediction studies. We apologize for the confusion caused by this mislabeling and will correct it in the revised paper for clarity.
>
> >Weakness: This works needs additional analysis or applications highlighting the unique advantage of the synthesized fMRI data from the proposed model.
>
> We greatly appreciate your thoughtful feedback and for highlighting the importance of discussing the practical implications and clinical relevance of our proposed method. In response to your concern, we have conducted three additional experiments.
>
> **First**, to validate the clinical applicability of our method, we conducted experiments on the ABIDE dataset. Considering that the HCP fMRI data used in our experiments are from healthy individuals, this experiment evaluates the generalizability of our method to fMRI data from individuals with certain neurological disorders, thereby demonstrating its potential for broader clinical applications. **Second**, to assess whether the generated fMRI data can support clinical applications, we evaluated their performance on downstream tasks, specifically cognitive prediction. **Finally**, to validate that the generated data can be reliably used for advanced brain analyses and further enhance their potential clinical utility, we evaluated whether our method retains brain network structures consistent with those observed in real brains through several physiological quantitative metrics.
>
> 1. Applicability to Disease-Related fMRI Data:
>
>     Our method is not only effective for fMRI data from healthy individuals but also demonstrates strong performance on data from individuals with certain diseases, such as those in the ABIDE dataset[2].
>
>     The ABIDE dataset compiles fMRI data from multiple international sites, focusing on brain imaging of individuals with Autism Spectrum Disorder (ASD) and Typically Developing Controls (TC). Data were collected across two phases: ABIDE I (2012) and ABIDE II (2016), with over 2,000 participants combined. The dataset includes resting-state fMRI scans from children, adolescents, and adults ages 7 to 64 years old, ensuring a broad demographic representation.
>
>     In the experiment, the context length is 32 and prediction length is 32. The results, as summarized in the table below, highlight the versatility of our method as a general-purpose fMRI generation approach. Our method's ability to generate high-quality fMRI signals for certain patients further underscores its application potential, reducing acquisition time and benefiting downstream tasks such as disease prediction.
>
>     | Method              | MAE $\downarrow$  | RMSE $\downarrow$ | MAPE $\downarrow$ |
>     |---------------------|---------|---------|--------|
>     | MSGNet[3] | 19.95   | 26.84   | 2.69   |
>     | U-Mixer[4]  | 18.74   | 24.97   | 1.55   |
>     | PDH-Diffusion (Ours)       | **18.15**   | **23.64**   | **1.48**   |
>
>     These findings show that our method achieves comparable performance to that of real fMRI data in downstream tasks. The results provide further insight into the effectiveness of our method in generating high-quality fMRI signals that are applicable to real-world tasks.  This reinforces the practical importance of our method in both clinical and research settings.
>
> [2] Heinsfeld, Anibal Sólon, et al. "Identification of autism spectrum disorder using deep learning and the ABIDE dataset." NeuroImage: Clinical 17 (2018): 16-23.
>
> [3] Cai, Wanlin, et al. "Msgnet: Learning multi-scale inter-series correlations for multivariate time series forecasting." Proceedings of the AAAI Conference on Artificial Intelligence. Vol. 38. No. 10. 2024.
>
> [4] Ma, Xiang, et al. "U-Mixer: An Unet-Mixer Architecture with Stationarity Correction for Time Series Forecasting." Proceedings of the AAAI Conference on Artificial Intelligence. Vol. 38. No. 13. 2024.

---

> ### Author Response · Authors · 2024-11-20
> **Response to Reviewer ouRH (Part 3)**
>
> 1. Applicability to Disease-Related fMRI Data:
>
>     Moreover, to intuitively illustrate the functional connectivity (FC) patterns between real and generated fMRI data for both Autism Spectrum Disorder (ASD) and Typically Developing Controls (TC), we visualized the FC in Figure 2 in anonymous link [https://anonymous.4open.science/r/iclr_6851/](https://anonymous.4open.science/r/iclr_6851/). In this visualization, green lines represent matching connections between the real and predicted FC, while red lines indicate discrepancies. As shown in the figure, our predicted data closely aligns with the real data in terms of FC patterns. Additionally, it is evident that individuals with ASD exhibit sparser FC connections compared to typically developing individuals. This underscores the potential of our generated fMRI data to provide reliable reference information for clinical diagnosis.
>
> 2. Evaluation on Downstream Tasks:
>
>     We further investigate the differences in downstream task performance when using fMRI signals generated by our proposed method compared to real fMRI data.
>
>     Predicting phenotypic characteristics using fMRI data is vital for inferring cognitive abilities, behavioral patterns, and other complex traits. This endeavor is crucial for deepening our understanding of the relationships between brain activity and behavior. It is especially significant in the study of psychiatric and developmental disorders, where variations in brain connectivity may be associated with cognitive or emotional functioning.
>
>     To validate the effectiveness of our predicted fMRI data in cognition prediction, we conducted experiments using the benchmark PTGB dataset[5]. We use real fMRI data for the training phase. For testing, we compare the results using real fMRI data and our predicted data with a prediction length of 64. Table below illustrates the performance of our method across five cognitive prediction tasks: **ReadEng**, **Flanker**, **PicSeq**, **ProcSpeed**, and **SCPT\_SEN**. Each cognitive task was evaluated using four standard metrics: Mean Absolute Error (MAE), Root Mean Square Error (RMSE), Mean Absolute Percentage Error (MAPE), and Correlation Coefficient (CC).
>
>     Table below shows the performance comparison of the downstream cognition prediction tasks using real and predicted fMRI signals.
>
>     | **Task** | **Metric** | **Real fMRI** | **Generated fMRI** |
>     | :-: | :-: | :-: | :-: |
>     | ReadEng | MAE  | 9.9580 | 10.8037 |
>     |  | RMSE | 14.2650 | 15.2069 |
>     |  | MAPE | 8.817  | 9.534 |
>     |  | CC   | 0.347  | 0.273 |
>     | Flanker | MAE  | 6.1559 | 6.333 |
>     |  | RMSE | 9.2298 | 8.979 |
>     |  | MAPE | 5.522  | 5.678 |
>     |  | CC   | 0.314  | 0.331 |
>     | PicSeq | MAE  | 11.4232 | 12.2050 |
>     |  | RMSE | 14.0081 | 15.0200 |
>     |  | MAPE | 11.071  | 11.803 |
>     |  | CC   | 0.333  | 0.290 |
>     | ProcSpeed | MAE  | 9.3325 | 9.4080 |
>     |  | RMSE | 11.3260 | 11.4040 |
>     |  | MAPE | 8.241  | 8.304 |
>     |  | CC   | 0.112  | 0.132 |
>     | SCPT_SEN | MAE  | 0.0708 | 0.0388 |
>     |  | RMSE | 0.1141 | 0.0720 |
>     |  | MAPE | 8.195  | 4.713 |
>     |  | CC   | 0.116  | 0.179 |
>
>     These findings show that our method achieves comparable performance to that of real fMRI data in downstream tasks. This further supports our motivation: by reducing acquisition time and cost, our approach enables the generation of high-quality fMRI data that retains its utility for critical applications, such as disease prediction and cognitive analysis. This reinforces the practical importance of our method in both clinical and research settings.
>
> [5] Yang, Yi, Hejie Cui, and Carl Yang. "Ptgb: Pre-train graph neural networks for brain network analysis." arXiv preprint arXiv:2305.14376 (2023).

---

> > ### Author Response · Authors · 2024-11-20
> > **Response to Reviewer ouRH (Part 4)**
> >
> > 3. Physiological Metrics of fMRI for Further Analysis:
> >
> >     To evaluate whether the fMRI signals generated by our model are consistent with real biological fMRI signals in terms of other neurological features, we employ three quantitative measures, following [6].
> >     First, we compute the Pearson correlation coefficients (PCCs) between the time series of resting-state fMRI signals from our generative model and those from the real brain. Second, we assess the similarity of the region-wise functional connectivity (FC) matrices derived from the generated and biological fMRI signals. This assessment involves calculating the PCCs between the two FC matrices and computing the Frobenius norm (F-norm) of the difference between them. The average PCC in the fMRI signals over all ROIs is **0.981**, the PCC between the FC matrices of our generative model and those from the real brain is **0.998**, and the F-norm distance between these FC matrices is **0.586**.
> >
> >     This validates the physiological alignment between the simulated fMRI signals and the biological brain data, demonstrating that the generated fMRI signals hold potential for broader applications in various brain analysis tasks.
> >
> > [6] Lu, Wenlian, et al. "Imitating and exploring the human brain's resting and task-performing states via brain computing: scaling and architecture." National Science Review 11.5 (2024): nwae080.

---

> ### Comment · Reviewer_ouRH · 2024-11-21
>
> Regarding Q1, thank you for your response.
>
> Perhaps, my question was not specific enough. I want to further clarify my question with more detailed concerns.
> My concern is related to validity of forecasting task. Predicting long timepoints is nearly impossible, especially when signal is pseudo-random such as fMRI signal. The authors even predicted 1-min timeseries (96 timepoints). To my view, the MAE error~30 (with 32 timepoints) simply suggest failure of prediction. That is why other baseline models yield comparable performance. This is also why trying to predict more samples (64 or 96 samples) only "slightly" increased the error.
>
> Below are my specific suggestions:
>
> 1. To examine whether predicted timepoints truly follow the actual dynamics of signal, please provide some examples (e.g., plot of forecasted vs. actual time series over 32 points).
>
> 2. To further validate the prediction of this extremely long periods is actually meaningful, please compare error to the simplest prediction model, straight-line, with statistical test.
>
> 3. Please calculate the error rate per timepoint and provide the trajectory of error over different length of prediction (1, 2, 3, ..., and up to 96). My speculation is that the error will reach to plateau at much earlier time points.
>
>
> This was my major concern why I put lower rate for this work. If the authors can prove that my above concern was wrong, I am willing to change my opinion regarding this work.

---

> > ### Author Response · Authors · 2024-11-26
> > **Response to Reviewer ouRH (Part 1)**
> >
> > Thank you for your valuable comments and for pointing out the detailed concerns regarding our experimental results. Below, we address your concerns point by point, providing additional details to clarify our results:
> >
> > > Question 1: To examine whether predicted timepoints truly follow the actual dynamics of signal, please provide some examples (e.g., plot of forecasted vs. actual time series over 32 points).
> >
> > To address this, we have plotted the results for 32 and 96 timepoints, as shown in Figure 3 (accessible via the anonymous link [https://anonymous.4open.science/r/iclr_6851/](https://anonymous.4open.science/r/iclr_6851/)). The figure includes two subplots, each corresponding to a different subject, with rows representing various ROIs. The plots demonstrate that the generated fMRI signals can follow the overall trends of the actual fMRI signals.
> >
> > We would also like to take this opportunity to clarify the motivation behind our work. The primary goal of our study is to generate fMRI signals that are as realistic as possible. Resting-state fMRI signals differ significantly from typical time series used in other studies, which leads to distinct research approaches and experimental outcomes. Physiologically, the fluctuation patterns of resting-state fMRI signals still have some aspects that remain unexplored and are yet to be fully understood[1]. This makes it extremely challenging to accurately extract detailed fluctuation features and trends, as is commonly done with conventional time series.
> >
> > Our comparative analysis with other baselines highlights this challenge, showing that existing time series prediction methods often struggle to predict fMRI signals effectively. Against this backdrop, our study represents a step forward in addressing this problem by incorporating brain physiological characteristics and dynamics. Considering that the value of fMRI data often lies in its downstream medical applications, such as disease diagnosis and cognitive prediction, preserving the physiological characteristics of real fMRI signals is crucial during generation and this is our objective of this work. Specifically, our goals are twofold:
> >
> > 1. To preserve the functional connectivity relationships between different brain regions (ROIs).
> > 2. To generate fMRI signals with dynamics characteristics that are closer to real physiological signals.
> >
> > At the same time, we also strive to minimize the prediction error. Compared to prior time series prediction methods, we believe our approach enables the generation of fMRI signals that are as realistic as possible, thereby facilitating downstream applications.
> >
> > While we acknowledge that perfectly replicating the detailed fluctuations of fMRI signals remains an ideal goal, achieving such precision is challenging, our work has addressed part of this challenge and, to some extent, improved the quality of the generated fMRI data. We hope our work will pave the way for future research to improve detailed fMRI prediction performance further.
> >
> > > Question 2: To further validate the prediction of this extremely long periods is actually meaningful, please compare error to the simplest prediction model, straight-line, with statistical test.
> >
> > Although we believe that the experimental results you mentioned are not directly related to the core contributions of our paper, we understand the importance of providing further clarification. In response to your suggestion, we have conducted the additional experiments as requested, with the hope of addressing any remaining concerns and offering a better understanding of our proposed method.
> >
> > We employed the simplest prediction model, linear regression, for fMRI prediction. The context length was set to 64, and the prediction length was set to 64. The results are as follows:
> >
> > | Method              | MAE $\downarrow$  | RMSE $\downarrow$ | MAPE (%)$\downarrow$ |
> > |---------------------|---------|---------|--------|
> > | Linear regression | 353.69   | 454.41   | 3.5681   |
> > | PDH-Diffusion (Ours)       | **29.04**   | **38.91**   | **0.2768**   |
> >
> > We are unsure whether we have fully understood your point. If not, we would greatly appreciate any further clarification of your question.
> >
> > [1] Schölvinck, Marieke L., et al. "Neural basis of global resting-state fMRI activity." Proceedings of the National Academy of Sciences 107.22 (2010): 10238-10243.

---

> ### Author Response · Authors · 2024-11-26
> **Response to Reviewer ouRH (Part 2)**
>
> > Question 3: Please calculate the error rate per timepoint and provide the trajectory of error over different length of prediction (1, 2, 3, ..., and up to 96). My speculation is that the error will reach to plateau at much earlier time points.
>
> Due to the time constraints associated with training a large number of models for all prediction lengths (1, 2, 3, ..., 96), we selected representative prediction lengths: 2, 16, 32, 48, 64, 80, and 96. The results are visualized in Figure 4 (accessible via the anonymous link [https://anonymous.4open.science/r/iclr_6851/](https://anonymous.4open.science/r/iclr_6851/)), where the error trajectory over these selected prediction lengths is presented.
>
> For this experimental result, we speculate that the model has already captured the key temporal dynamics within the provided context, making additional prediction steps less impactful. Since resting-state fMRI time series exhibit relatively stable dynamic patterns[1][2], the information entropy of the fMRI data is limited. Even when predicting longer time steps, the underlying changes in the data may not vary significantly. If the context length of 64 already provides sufficient information to capture the brain activity patterns, the model has likely learned the essential dynamic relationships within this shorter context. As a result, varying the model's prediction length does not lead to significant changes in performance.
>
> > Weakness: My concern is related to validity of forecasting task. Predicting long timepoints is nearly impossible, especially when signal is pseudo-random such as fMRI signal.
>
> We appreciate your concern regarding the difficulty of forecasting long timepoints, especially for signals like fMRI that appear pseudo-random. However, we would like to clarify that the fMRI signals are not completely random as they reflect the interactions and activation patterns between different brain regions. These signals exhibit certain regularities and inherent correlations between each other, which make prediction feasible. Currently, there are studies on predicting brain signals, such as EEG signals[3][4], which demonstrate that despite the inherent randomness of brain signals, it is still possible to effectively capture their trends and patterns. Furthermore, there have been studies in other fields that focus on predicting and generating signals with fractal characteristics, such as [5][6]. This demonstrates that time series with pseudo-random properties also hold meaningful potential for long-term sequence generation. While predicting long-term fMRI time series presents challenges, it remains meaningful in certain application scenarios, such as improving the efficiency of fMRI data acquisition or simulating long-term brain activity.
>
> > Weakness: To my view, the MAE error~30 (with 32 timepoints) simply suggest failure of prediction. That is why other baseline models yield comparable performance. This is also why trying to predict more samples (64 or 96 samples) only "slightly" increased the error.
>
> Thank you for your observation regarding the MAE error. We would like to clarify that while the MAE value of ~30 may seem large at first glance, it is important to consider the magnitude of fMRI signal values, which are inherently large (approximately 10000). In this context, the MAPE (Mean Absolute Percentage Error) is quite small. For example, on the HCP dataset, when the context length is 64 and the prediction length is 32, the MAPE is only 0.2744%. When the prediction length is increased to 96, the MAPE remains similarly small at 0.2803%. This indicates that our model has captured the overall trends and distribution of the fMRI data effectively.
>
>
> [1] Thomason, Moriah E., et al. "Resting-state fMRI can reliably map neural networks in children." Neuroimage 55.1 (2011): 165-175.
>
> [2] Patriat, Rémi, et al. "The effect of resting condition on resting-state fMRI reliability and consistency: a comparison between resting with eyes open, closed, and fixated." Neuroimage 78 (2013): 463-473.
>
> [3] Nguyen, Phuong Thi Mai, et al. "Collective almost synchronization-based model to extract and predict features of EEG signals." Scientific reports 10.1 (2020): 16342.
>
> [4] Shen, Fang, Jing Liu, and Kai Wu. "Multivariate time series forecasting based on elastic net and high-order fuzzy cognitive maps: a case study on human action prediction through EEG signals." IEEE Transactions on Fuzzy Systems 29.8 (2020): 2336-2348.
>
> [5] Chen, Hua, et al. "Spatio-temporal graph attention network for sintering temperature long-range forecasting in rotary kilns." IEEE Transactions on Industrial Informatics 19.2 (2022): 1923-1932.
>
> [6] Coulibaly, Paulin, and Connely K. Baldwin. "Nonstationary hydrological time series forecasting using nonlinear dynamic methods." Journal of Hydrology 307.1-4 (2005): 164-174.

---

> > ### Comment · Reviewer_ouRH · 2024-11-26
> >
> > Thank you for your effort on resolving my concerns.
> > Although I have some concerns; I strongly believe the merit of this work clearly outweighs weakness or potential concerns of this work.
> >
> > I am going to increase my score. Just try to include plot of original vs. predicted fMRI timeseries in the revised manuscript.

---

> > > ### Author Response · Authors · 2024-11-27
> > > **Response to Reviewer ouRH**
> > >
> > > We have updated the revised version of the manuscript, incorporating your valuable feedback. We would like to sincerely thank you for your support of our work. Your constructive suggestions have been incredibly helpful in improving the quality of our manuscript.

---

### Official Review · Reviewer_fBZF · 2024-11-01

**Soundness:** 3
**Presentation:** 4
**Contribution:** 4
**Rating:** 6
**Confidence:** 4

**Summary:**

This paper proposes 'PDH-Diffusion', a framework that synthesizes fMRI data through a hypergraph-based hierarchical signals generator, the properties of a dynamic guiding module, and a cross-brain region guiding progressive diffusion model. The authors first introduced scientific priors, motivation, and concrete realization for building these modules in detail. They then provided the quantified results compared with previous approaches and the ablation experiment, showing their architecture has better performance in generating fMRI than other baseline models.

**Strengths:**

1. The paper established a novel framework for fMRI synthesis taking the dynamic characteristics and structure of the brain into consideration which has the solid and strong theoretical support.
2. The model incorporates hierarchical brain regional interactions and multifractal dynamics, enabling it to generate fMRI signals that better reflect physiological properties.

**Weaknesses:**

1. In the ''Experimental Setups'' section, the experiment did not test on multiple fMRI datasets or finer parcellation atlas, but only used the HCP dataset on the AAL atlas, which makes the results of the experiment not that solid. Meanwhile, I would suggest the framework could be extended to and tested in other modalities, for instance, EEG signals.
2. Although the paper introduces a novel framework for generating fMRI data, it lacks validation on the generated data’s impact on downstream tasks, such as disease diagnosis, or brain network analysis.

**Questions:**

1. As noted in the weaknesses, synthetic neural signals are particularly intriguing in the neuroscience field. Would this framework be adaptable for application or testing on other fMRI datasets, especially those with different clinical populations, to assess its generalizability?

2. Typically, fMRI signals are recorded in a low temporal resolution. Could the model be extended to work with high temporal resolution modalities, such as magnetoencephalography (MEG) or electroencephalography (EEG), where realistic signal synthesis could enhance downstream analyses?

---

> ### Author Response · Authors · 2024-11-20
> **Response to Reviewer fBZF (Part 1)**
>
> Thank you very much to your careful review. Below, we provide responses to your questions and address your concerns one by one. We hope these clarifications resolve your doubts.
>
> > Question1: As noted in the weaknesses, synthetic neural signals are particularly intriguing in the neuroscience field. Would this framework be adaptable for application or testing on other fMRI datasets, especially those with different clinical populations, to assess its generalizability?
>
> Thank you for your valuable suggestion. We have extended our experiments to include the ABIDE dataset[3], which comprises fMRI data from individuals with Autism Spectrum Disorder (ASD). The ABIDE dataset compiles fMRI data from multiple international sites, focusing on brain imaging of individuals with autism spectrum disorder (ASD) and typically developing controls (TC). Data were collected across two phases: ABIDE I (2012) and ABIDE II (2016), with over 2,000 participants combined. The dataset includes resting-state fMRI scans from children, adolescents, and adults ages 7 to 64 years old, ensuring a broad demographic representation.
>
> In the experiment, the context length is 32 and prediction length is 32. The results, as summarized in the table below, highlight the versatility of our method as a general-purpose fMRI generation approach. Our method's ability to generate high-quality fMRI signals for certain patients further underscores its application potential, reducing acquisition time and benefiting downstream tasks such as disease prediction.
>
>
> | Method              | MAE $\downarrow$  | RMSE $\downarrow$ | MAPE $\downarrow$ |
> |---------------------|---------|---------|--------|
> | MSGNet[1] | 19.95   | 26.84   | 2.69   |
> | U-Mixer[2]  | 18.74   | 24.97   | 1.55   |
> | PDH-Diffusion (Ours)       | **18.15**   | **23.64**   | **1.48**   |
>
> These findings show that our method achieves comparable performance to that of real fMRI data in downstream tasks. The results provide further insight into the effectiveness of our method in generating high-quality fMRI signals that are applicable to real-world tasks.  This reinforces the practical importance of our method in both clinical and research settings.
>
> Moreover, to intuitively illustrate the functional connectivity (FC) patterns between real and generated fMRI data for both Autism Spectrum Disorder (ASD) and Typically Developing Controls (TC), we visualized the FC in Figure 2 in anonymous link [https://anonymous.4open.science/r/iclr_6851/](https://anonymous.4open.science/r/iclr_6851/). In this visualization, green lines represent matching connections between the real and predicted FC, while red lines indicate discrepancies. As shown in the figure, our predicted data closely aligns with the real data in terms of FC patterns. Additionally, it is evident that individuals with ASD exhibit sparser FC connections compared to typically developing individuals. This underscores the potential of our generated fMRI data to provide reliable reference information for clinical diagnosis.
>
> [1] Cai, Wanlin, et al. "Msgnet: Learning multi-scale inter-series correlations for multivariate time series forecasting." Proceedings of the AAAI Conference on Artificial Intelligence. Vol. 38. No. 10. 2024.
>
> [2] Ma, Xiang, et al. "U-Mixer: An Unet-Mixer Architecture with Stationarity Correction for Time Series Forecasting." Proceedings of the AAAI Conference on Artificial Intelligence. Vol. 38. No. 13. 2024.
>
> [3] Heinsfeld, Anibal Sólon, et al. "Identification of autism spectrum disorder using deep learning and the ABIDE dataset." NeuroImage: Clinical 17 (2018): 16-23.

---

> ### Author Response · Authors · 2024-11-20
> **Response to Reviewer fBZF (Part 2)**
>
> > Question2: Typically, fMRI signals are recorded in a low temporal resolution. Could the model be extended to work with high temporal resolution modalities, such as magnetoencephalography (MEG) or electroencephalography (EEG), where realistic signal synthesis could enhance downstream analyses?
>
> Since EEG signals also exhibit multifractal characteristics[4][5] and contain information about brain structure, our method can theoretically be applied to the generation of EEG signals as well. However, as EEG data often requires specific permissions and access, this extension would require additional time and resources.
>
> In this paper, we focus on generating fMRI signals, primarily due to their low temporal resolution, which limits their ability to capture dynamic neural processes. Despite their high spatial resolution, which makes them indispensable for tasks requiring precise localization of neural activity, fMRI data face several challenges, including the need for stringent experimental conditions where subjects must remain completely still during scans and the high costs of data acquisition. By addressing the temporal resolution limitation through generative modeling, we aim to enhance the utility of fMRI data for downstream applications.
>
> On the other hand, EEG signals inherently possess high temporal resolution, making them ideal for capturing rapid neural dynamics. However, their primary drawback is low spatial resolution, which limits their effectiveness in tasks that demand precise spatial localization of brain activity. For EEG data, augmenting the spatial dimension through generative models would be a more impactful approach to improve their applicability in downstream tasks. This remains an interesting avenue for future exploration. We plan to investigate this direction in our future work, as it aligns with our broader goal of improving data synthesis techniques for diverse neuroimaging modalities.
>
> > Weakness1: In the ''Experimental Setups'' section, the experiment did not test on multiple fMRI datasets or finer parcellation atlas, but only used the HCP dataset on the AAL atlas, which makes the results of the experiment not that solid. Meanwhile, I would suggest the framework could be extended to and tested in other modalities, for instance, EEG signals.
>
> This part of the response is the same as our response to Question 2.
>
> > Weakness2: Although the paper introduces a novel framework for generating fMRI data, it lacks validation on the generated data’s impact on downstream tasks, such as disease diagnosis, or brain network analysis.
>
> Thank you for your valuable suggestion. To address your concern and evaluate the performance of our generated signals on downstream tasks, we have added two additional experiments. **The first experiment** aims to verify that the data generated by our method retain brain network structures consistent with those of real brains, enabling their use in more advanced brain analyses. To achieve this, we evaluated several physiological quantitative metrics. **The second experiment** focuses on assessing whether the generated data possess the same clinical applicability as real fMRI data. Specifically, we evaluated the performance of the generated data on downstream tasks such as cognitive prediction.
>
>
> [4] Lutzenberger, Werner, et al. "The scalp distribution of the fractal dimension of the EEG and its variation with mental tasks." Brain topography 5.1 (1992): 27-34.
>
> [5] Kesić, Srdjan, and Sladjana Z. Spasić. "Application of Higuchi's fractal dimension from basic to clinical neurophysiology: a review." Computer methods and programs in biomedicine 133 (2016): 55-70.

---

> ### Author Response · Authors · 2024-11-20
> **Response to Reviewer fBZF (Part 3)**
>
> Firstly, we investigate the differences in cognitive prediction task performance when using fMRI signals generated by our proposed method compared to real fMRI data. Predicting phenotypic characteristics using fMRI data is vital for inferring cognitive abilities, behavioral patterns, and other complex traits. This endeavor is crucial for deepening our understanding of the relationships between brain activity and behavior. It is especially significant in the study of psychiatric and developmental disorders, where variations in brain connectivity may be associated with cognitive or emotional functioning.
>
> To validate the effectiveness of our predicted fMRI data in cognition prediction, we conducted experiments using the benchmark PTGB dataset[6]. We use real fMRI data for the training phase. For testing, we compare the results using real fMRI data and our predicted data with a prediction length of 64. Table below illustrates the performance of our method across five cognitive prediction tasks: **ReadEng**, **Flanker**, **PicSeq**, **ProcSpeed**, and **SCPT\_SEN**. Each cognitive task was evaluated using four standard metrics: Mean Absolute Error (MAE), Root Mean Square Error (RMSE), Mean Absolute Percentage Error (MAPE), and Correlation Coefficient (CC).
>
> Table below shows the performance comparison of the downstream cognition prediction tasks using real and predicted fMRI signals.
>
> | **Task** | **Metric** | **Real fMRI** | **Generated fMRI** |
> | :-: | :-: | :-: | :-: |
> | ReadEng | MAE  | 9.9580 | 10.8037 |
> |  | RMSE | 14.2650 | 15.2069 |
> |  | MAPE | 8.817  | 9.534 |
> |  | CC   | 0.347  | 0.273 |
> | Flanker | MAE  | 6.1559 | 6.333 |
> |  | RMSE | 9.2298 | 8.979 |
> |  | MAPE | 5.522  | 5.678 |
> |  | CC   | 0.314  | 0.331 |
> | PicSeq | MAE  | 11.4232 | 12.2050 |
> |  | RMSE | 14.0081 | 15.0200 |
> |  | MAPE | 11.071  | 11.803 |
> |  | CC   | 0.333  | 0.290 |
> | ProcSpeed | MAE  | 9.3325 | 9.4080 |
> |  | RMSE | 11.3260 | 11.4040 |
> |  | MAPE | 8.241  | 8.304 |
> |  | CC   | 0.112  | 0.132 |
> | SCPT_SEN | MAE  | 0.0708 | 0.0388 |
> |  | RMSE | 0.1141 | 0.0720 |
> |  | MAPE | 8.195  | 4.713 |
> |  | CC   | 0.116  | 0.179 |
>
>
> These findings show that our method achieves comparable performance to that of real fMRI data in downstream tasks. This further supports our motivation: by reducing acquisition time and cost, our approach enables the generation of high-quality fMRI data that retains its utility for critical applications, such as disease prediction and cognitive analysis. This reinforces the practical importance of our method in both clinical and research settings.
>
> Secondly, to evaluate whether the fMRI signals generated by our model are consistent with real biological fMRI signals in terms of other neurological features, we employ three quantitative measures, following [7].
> First, we compute the Pearson correlation coefficients (PCCs) between the time series of resting-state fMRI signals from our generative model and those from the real brain. Second, we assess the similarity of the region-wise functional connectivity (FC) matrices derived from the generated and biological fMRI signals. This assessment involves calculating the PCCs between the two FC matrices and computing the Frobenius norm (F-norm) of the difference between them. The average PCC in the fMRI signals over all ROIs is **0.981**, the PCC between the FC matrices of our generative model and those from the real brain is **0.998**, and the F-norm distance between these FC matrices is **0.586**.
>
> This validates the physiological alignment between the simulated fMRI signals and the biological brain data, demonstrating that the generated fMRI signals hold potential for broader applications in various brain analysis tasks.
>
> [6] Yang, Yi, Hejie Cui, and Carl Yang. "Ptgb: Pre-train graph neural networks for brain network analysis." arXiv preprint arXiv:2305.14376 (2023).
>
> [7] Lu, Wenlian, et al. "Imitating and exploring the human brain's resting and task-performing states via brain computing: scaling and architecture." National Science Review 11.5 (2024): nwae080.

---

> > ### Comment · Reviewer_fBZF · 2024-11-23
> >
> > Thanks for your reply. I have decided to change my rating to 6.
> > It is an interesting study, I hope that the proposed technique can really benefit the neuroscience field.

---

> > > ### Author Response · Authors · 2024-11-27
> > > **Response to Reviewer fBZF**
> > >
> > > We are truly grateful for your valuable feedback and assistance you've offered. We also sincerely appreciate your support for our work. Thank you for your kind consideration in increasing the score!

---

### Official Review · Reviewer_hv4T · 2024-11-02

**Soundness:** 3
**Presentation:** 3
**Contribution:** 2
**Rating:** 5
**Confidence:** 4

**Summary:**

In this paper, the authors proposed a novel framework named the Physiological Dynamics-Driven Hierarchical Diffusion Model (PDH-Diffusion) for fMRI analytics. The PDH-Diffusion framework integrates two essential brain physiological properties, hierarchical regional interactions and multifractal dynamics, into the diffusion process. The primary goal is to improve diffusion models’ capability to generate realistic fMRI time series signals by accurately capturing these physiological characteristics.

**Strengths:**

Overall, the major strength of this work lies in its novelty. The authors have developed an innovative framework that captures complex interdependencies and multifractal dynamics within synthetic fMRI signals.

Specifically, their contribution includes integrating three key components into the diffusion process: a hypergraph-based signal generator, a dynamics properties guiding module, and a cross-brain region progressive diffusion model. This integration enhances the realism of the generated signals. The authors provide a robust theoretical foundation for their methods and perform extensive quantitative analysis, demonstrating the framework’s accuracy and effectiveness in time series forecasting. The paper is well-organized and includes relevant background information. Results from the proposed method outperform multiple peer models in time-series forecasting and diffusion models, as evidenced by superior MAE, MAPE, and RMSE scores, highlighting the model’s effectiveness.

**Weaknesses:**

The reviewers have multiple concerns about the framework and potential impact in this work.

1). The confusion about physiological fMRI. Usually, fMRI are categorized into resting-state and task-based fMRI. The resting-state fMRI is commonly scanned without specific stimulus, whereas task-based fMRI is acquired based on external stimulus, such as 7 tasks in HCP. Is physiological fMRI is either resting-state or task-based signal? The authors do not clarify the concept even in Introdcution section.

2). Limited motivation and impact. In Abstract, the authors mentioned "Functional magnetic resonance imaging (fMRI) is essential for mapping brain activity but faces challenges like lengthy acquisition time and sensitivity to patient movement, limiting its clinical and machine learning applications." It seems that the authros' work can advance the fMRI for clincial application, but the authors do not generate some neurological or psychiatric fMRI to validate. From reviewers' perspective, using the innovative PDH-Diffusion model, it can assist physician to provide lengthy fMRI signal which will denfinitely reduce the inconvenience of patients. Only generating healthy fMRI can impair the motivation and impact of this work.

3). Lacking of qualitative comparison. The authors have provided an extensive quantitative validation of PDH-Diffusion model with other peer methods using MAE, MAPE, and RMSE. Unfortunately, the autors do not provide any qualitative results, such as Functional Connectivity Map or Brain Connectivity Maps, of PDH-Diffusion. That is, although averaging metrics such as MAE, RMSE, MAPE across 10 runs may demonstrate robustness, these metrics  cannot fully capture the quality or realism of the synthesized fMRI signals. Notably, the qualitative results is also vital in clinics, since these results showcase which brain regions are severly impaired by neurological disorders.  Importantly, there is no visual representation given of the generated fMRI signal, which would be valuable for assessing their plausibility.

4). Several technical issues. The variance schedule (parameters $\alpha_n$ and $\beta_n$) in the diffusion process may not be fully optimized for different regions or scales, potentially leading to inappropriate noise levels in certain hierarchical levels. This could result in over-smoothing or overfitting at certain levels. Additionally, conditioning on historical data could lead to overfitting if the model becomes too dependent on past values, especially if the training data does not represent the full spectrum of brain dynamics. Without explicit mitigation measures, such as adaptive variance schedules or regularization techniques, these issues may limit the model’s ability to generalize to new or varied patterns, impacting its robustness and effectiveness.

5). Multiple typographical mistakes. In Section 4.3 (ABLATION ANALYSIS) there are many typographical errors. The term “share radio” is used instead of ‘share ratio’ in “The influence of share radio” this section. Similar typographical errors appear in the caption and text of Figure 3(b), as well as in Section 3.4. These typographical mistakes impair the redability of this paper.

**Questions:**

The reviewers have raised several questions regarding the weaknesses of this work:

1). Qualitative Comparisons: The authors are strongly encouraged to provide qualitative comparisons between the synthesized and real fMRI signals to allow for a qualitative assessment of the model’s performance.

2). Optimal Range of Brain Regions: The results suggest an “optimal” range of brain regions that enhances performance. How is this range determined, and is it manually set? Reviewers are concerned about the reliance on manual design for determining this optimal range.

3). Validation of Multifractal Properties: How do the authors validate that the generated signals preserve multifractal properties?

4). Risk of Overfitting: Do the variance schedule parameters ($\alpha_n$ and $\beta_n$) and the historical data used in training the PDH-Diffusion model lead to overfitting?

---

> ### Author Response · Authors · 2024-11-20
> **Response to Reviewer hv4T (Part 1)**
>
> Thank you for your detailed review and insightful suggestions. Below, we provide responses to your questions. We hope these clarifications resolve your doubts.
>
> > Question1: Qualitative Comparisons: The authors are strongly encouraged to provide qualitative comparisons between the synthesized and real fMRI signals to allow for a qualitative assessment of the model’s performance.
>
> To better evaluate the fMRI data generated by our method, we have added two additional experiments. **The first experiment** aims to verify that the data generated by our method retain brain network structures consistent with those of real brains, enabling their use in more advanced brain analyses. To achieve this, we evaluated several physiological quantitative metrics. **The second experiment** focuses on assessing whether the generated data possess the same clinical applicability as real fMRI data. Specifically, we evaluated the performance of the generated data on downstream tasks such as cognitive prediction.
>
> To assess the consistency between the fMRI signals generated by our method and real biological fMRI signals in terms of neurological characteristics, we utilize three quantitative metrics, following[1].
>
> First, we compute the Pearson correlation coefficients (PCCs) between the time series of resting-state fMRI signals from our generative model and those from the real brain. Second, we assess the similarity of the region-wise functional connectivity (FC) matrices derived from the generated and biological fMRI signals. This assessment involves calculating the PCCs between the two FC matrices and computing the Frobenius norm (F-norm) of the difference between them. The average PCC in the fMRI signals over all ROIs is **0.981**, the PCC between the FC matrices of our generative model and those from the real brain is **0.998**, and the F-norm distance between these FC matrices is **0.586**.
>
> This validates the physiological alignment between the simulated fMRI signals and the biological brain data, demonstrating that the generated fMRI signals hold potential for broader applications in various brain analysis tasks.
>
> In addition, we evaluated downstream tasks of fMRI data, focusing on cognitive prediction. These evaluations were conducted on both synthesized and real fMRI signals, enabling a direct comparison of their performance in the task. Predicting phenotypic characteristics using fMRI data is vital for inferring cognitive abilities, behavioral patterns, and other complex traits. This endeavor is crucial for deepening our understanding of the relationships between brain activity and behavior.
>
> To validate the effectiveness of our predicted fMRI data in cognition prediction, we conducted experiments using the benchmark PTGB dataset[2]. We use real fMRI data for the training phase. For testing, we compare the results using real fMRI data and our predicted data with a prediction length of 64. Table below illustrates the performance of our method across five cognitive prediction tasks: **ReadEng**, **Flanker**, **PicSeq**, **ProcSpeed**, and **SCPT_SEN**. Each cognitive task was evaluated using four standard metrics: Mean Absolute Error (MAE), Root Mean Square Error (RMSE), Mean Absolute Percentage Error (MAPE), and Correlation Coefficient (CC).
>
> Table below shows the performance comparison of the downstream cognition prediction tasks using real and predicted fMRI signals.
>
> | **Task** | **Metric** | **Real fMRI** | **Generated fMRI** |
> | :-: | :-: | :-: | :-: |
> | ReadEng | MAE  | 9.9580 | 10.8037 |
> |  | RMSE | 14.2650 | 15.2069 |
> |  | MAPE | 8.817  | 9.534 |
> |  | CC   | 0.347  | 0.273 |
> | Flanker | MAE  | 6.1559 | 6.333 |
> |  | RMSE | 9.2298 | 8.979 |
> |  | MAPE | 5.522  | 5.678 |
> |  | CC   | 0.314  | 0.331 |
> | PicSeq | MAE  | 11.4232 | 12.2050 |
> |  | RMSE | 14.0081 | 15.0200 |
> |  | MAPE | 11.071  | 11.803 |
> |  | CC   | 0.333  | 0.290 |
> | ProcSpeed | MAE  | 9.3325 | 9.4080 |
> |  | RMSE | 11.3260 | 11.4040 |
> |  | MAPE | 8.241  | 8.304 |
> |  | CC   | 0.112  | 0.132 |
> | SCPT_SEN | MAE  | 0.0708 | 0.0388 |
> |  | RMSE | 0.1141 | 0.0720 |
> |  | MAPE | 8.195  | 4.713 |
> |  | CC   | 0.116  | 0.179 |
>
> These findings show that our method achieves comparable performance to that of real fMRI data in downstream tasks. This further supports our motivation: by reducing acquisition time and cost, our approach enables the generation of high-quality fMRI data that retains its utility for critical applications, such as disease prediction and cognitive analysis. This reinforces the practical importance of our method in both clinical and research settings.
>
> [1] Lu, Wenlian, et al. "Imitating and exploring the human brain's resting and task-performing states via brain computing: scaling and architecture." National Science Review 11.5 (2024): nwae080.
>
> [2] Yang, Yi, Hejie Cui, and Carl Yang. "Ptgb: Pre-train graph neural networks for brain network analysis." arXiv preprint arXiv:2305.14376 (2023).

---

> ### Author Response · Authors · 2024-11-20
> **Response to Reviewer hv4T (Part 2)**
>
> > Question2: Optimal Range of Brain Regions: The results suggest an “optimal” range of brain regions that enhances performance. How is this range determined, and is it manually set? Reviewers are concerned about the reliance on manual design for determining this optimal range.
>
> In our experiments, we partitioned the brain regions by selecting the number of neighbors for each region based on the magnitude of Pearson correlation coefficients among brain ROIs and it is manually set. We acknowledge your concern regarding the reliance on manual design in determining the optimal range. However, Pearson correlation is commonly used in medical research to analyze and define functional boundaries of brain regions[3][4]. Using Pearson correlation to define functional ranges of brain regions is a widely accepted practice in neuroimaging studies.
>
> > Question3: Validation of Multifractal Properties: How do the authors validate that the generated signals preserve multifractal properties?
>
> Thank you for your insightful question regarding the validation of the multifractal properties of the generated signals. To ensure that our generated signals accurately preserve the multifractal characteristics inherent in brain signals, we computed the the $q$-order singularity exponent $hq$ and corresponding dimension $Dq$ for both the generated signals and the real fMRI signals. Additionally, we compared our proposed method with other state-of-the-art prediction methods (MSGNet[5], U-Mixer[6]) in terms of fractal characteristics. The results, including those for real fMRI data, are plotted as functions of the order $q$ in Figure 1 in anonymous link [https://anonymous.4open.science/r/iclr_6851/](https://anonymous.4open.science/r/iclr_6851/).  The results are averaged over all test subjects. and demonstrate that the multifractal characteristics of the fMRI data generated by our method is closer to that of the real fMRI data compared to those generated by other methods. This analysis confirms that our method effectively maintains the multifractal properties of the original brain signals.
>
> > Question4: Risk of Overfitting: Do the variance schedule parameters ($\alpha_n$ and $\beta_n$) and the historical data used in training the PDH-Diffusion model lead to overfitting?
>
> Thank you for raising this important question. We believe that the variance schedule parameters and the historical data used in training the PDH-Diffusion model do not lead to overfitting. This is because we divided the training and testing datasets by individuals, ensuring that no data from the individuals in the test set was seen during training. This partitioning approach inherently demonstrates the cross-individual generalization ability of our method during testing. If we have misunderstood your concern, please let us know. We would be more than happy to provide further clarification and address your questions in more detail.
>
> > Weakness1: The confusion about physiological fMRI. Usually, fMRI are categorized into resting-state and task-based fMRI. The resting-state fMRI is commonly scanned without specific stimulus, whereas task-based fMRI is acquired based on external stimulus, such as 7 tasks in HCP. Is physiological fMRI is either resting-state or task-based signal? The authors do not clarify the concept even in Introdcution section.
>
> We apologize for the confusion caused by the lack of detailed descriptions in the paper. The data we used in experiments is resting-state fMRI from HCP[8]. For fMRI data in HCP dataset, the repetition time (TR) is 720 $\text{ms}$ and the echo time (TE) is 33.1 $\text{ms}$. The voxel size is 2$\text{mm}^3$.
>
> [3] Li, Weikai, et al. "Remodeling Pearson's correlation for functional brain network estimation and autism spectrum disorder identification." Frontiers in neuroinformatics 11 (2017): 55.
>
> [4] Segall, Judith M., et al. "Correspondence between structure and function in the human brain at rest." Frontiers in neuroinformatics 6 (2012): 10.
>
> [5] Cai, Wanlin, et al. "Msgnet: Learning multi-scale inter-series correlations for multivariate time series forecasting." Proceedings of the AAAI Conference on Artificial Intelligence. Vol. 38. No. 10. 2024.
>
> [6] Ma, Xiang, et al. "U-Mixer: An Unet-Mixer Architecture with Stationarity Correction for Time Series Forecasting." Proceedings of the AAAI Conference on Artificial Intelligence. Vol. 38. No. 13. 2024.

---

> ### Author Response · Authors · 2024-11-20
> **Response to Reviewer hv4T (Part 3)**
>
> > Weakness2: It seems that the authros' work can advance the fMRI for clincial application, but the authors do not generate some neurological or psychiatric fMRI to validate.
>
> To validate the clinical applicability of our method, we conducted experiments on the ABIDE dataset[7]. Considering that the HCP fMRI data used in our experiments are from healthy individuals, we evaluated the generalizability of our method to fMRI data from individuals with certain neurological disorders, thereby demonstrating its potential for broader clinical applications.     The ABIDE dataset compiles fMRI data from multiple international sites, focusing on brain imaging of individuals with Autism Spectrum Disorder (ASD) and Typically Developing Controls (TC). Data were collected across two phases: ABIDE I (2012) and ABIDE II (2016), with over 2,000 participants combined. The dataset includes resting-state fMRI scans from children, adolescents, and adults ages 7 to 64 years old, ensuring a broad demographic representation.
>
> In the experiment, the context length is 32 and prediction length is 32. The results, as summarized in the table below, highlight the versatility of our method as a general-purpose fMRI generation approach. Our method's ability to generate high-quality fMRI signals for certain patients further underscores its application potential, reducing acquisition time and benefiting downstream tasks such as disease prediction.
>
> | Method              | MAE $\downarrow$  | RMSE $\downarrow$ | MAPE $\downarrow$ |
> |---------------------|---------|---------|--------|
> | MSGNet[5] | 19.95   | 26.84   | 2.69   |
> | U-Mixer[6]  | 18.74   | 24.97   | 1.55   |
> | PDH-Diffusion (Ours)       | **18.15**   | **23.64**   | **1.48**   |
>
> These findings show that our method achieves comparable performance to that of real fMRI data in downstream tasks. The results provide further insight into the effectiveness of our method in generating high-quality fMRI signals that are applicable to real-world tasks.  This reinforces the practical importance of our method in both clinical and research settings.
>
> Moreover, to intuitively illustrate the functional connectivity (FC) patterns between real and generated fMRI data for both Autism Spectrum Disorder (ASD) and Typically Developing Controls (TC), we visualized the FC in Figure 2 in anonymous link [https://anonymous.4open.science/r/iclr_6851/](https://anonymous.4open.science/r/iclr_6851/). In this visualization, green lines represent matching connections between the real and predicted FC, while red lines indicate discrepancies. As shown in the figure, our predicted data closely aligns with the real data in terms of FC patterns. Additionally, it is evident that individuals with ASD exhibit sparser FC connections compared to typically developing individuals. This underscores the potential of our generated fMRI data to provide reliable reference information for clinical diagnosis.
>
> >Weakness3: Lacking of qualitative comparison.
>
> As addressed in our response to Question 1, we add qualitative comparisons which demonstrates the clinical effectiveness of our proposed method.
>
> >Weakness4: Several technical issues.
>
> This part of the response is the same as our response to Question 4.
>
> >Weakness5: Typographical mistakes.
>
> We sincerely apologize for the typing error, and we greatly appreciate your pointing it out. We will make corrections in the revised paper.
>
> [8] Van Essen, David C., et al. "The Human Connectome Project: a data acquisition perspective." Neuroimage 62.4 (2012): 2222-2231.

---

> > ### Comment · Reviewer_hv4T · 2024-12-02
> >
> > Thanks to the authors for clarifying! Indeed, one of the most significant advantages of fMRI is its capability to highlight specific brain regions affected by neurological disorders. However, even with the augmentation of the original fMRI signal, precisely identifying the brain regions associated with these disorders remains a formidable challenge. For instance, while successfully enlarging the original fMRI signal from 100 × 262,309 to 1,200 × 262,309 might provide more information and improve prediction accuracy, clinical physicians may still prefer EEG over fMRI due to its practicality and accessibility. Nonetheless, if the methodology proposed in this manuscript facilitates the identification of innovative spatial information (e.g., spatial features), it could mark a significant advancement in fMRI analytics. Such progress would undoubtedly push the boundaries of fMRI research and pave the way for transformative clinical translational applications.
> >
> > While the technology introduced in this paper is promising, it still falls short in revealing more detailed and sophisticated spatial features for direct clinical implementation. As such, I will maintain the current score for this manuscript. Additionally, I recommend the following important reference to enhance the discussion of spatial features within fMRI analytics:
> >
> > Smith, S. M., Fox, P. T., Miller, K. L., Glahn, D. C., Fox, P. M., Mackay, C. E., ... & Beckmann, C. F. (2009). Correspondence of the brain's functional architecture during activation and rest. Proceedings of the National Academy of Sciences, 106:13040-13045.

---

> > > ### Author Response · Authors · 2024-12-03
> > > **Response to Reviewer hv4T**
> > >
> > > Thank you for your thoughtful feedback and for acknowledging the value of our work. Compared to EEG signals, fMRI signals have the significant advantages of higher spatial resolution and the ability to capture deep brain activity. While we acknowledge that clinical physicians may prefer EEG over fMRI, this preference is primarily attributed to technical limitations, such as the high acquisition cost and the time-consuming scanning process of fMRI. These limitations have indeed motivated our research: by generating more realistic fMRI signals, we aim to facilitate the broader clinical application of fMRI.
> > >
> > > Your suggestion to emphasize the identification of innovative spatial information (e.g., spatial features) is insightful and aligns well with the objectives of our work. The fMRI signals generated by our method can reflect brain activation patterns to some extent, thereby offering insights into brain states and potential disease locations. As we discussed in our response to Weakness2 ( in "Response to Reviewer hv4T (Part 3)" ), the sparsity of functional connectivity between disease and health brain states is different, which reflects the potential to provide reliable reference information for clinical diagnosis. Additionally, based on our generated fMRI data, we further visualized brain activation maps for patients with neurological disorders and healthy controls, which are shown in the figure 5 in anonymous link [https://anonymous.4open.science/r/iclr_6851/](https://anonymous.4open.science/r/iclr_6851/). These visualizations demonstrate that the generated fMRI signals can effectively reflect brain activation states, providing valuable insights for more downstream research.
> > >
> > > Given that the primary focus of this work is on the design of the generation algorithm guided by physiological characteristics, we conducted extensive experiments to evaluate the accuracy of our generated fMRI signals in the initial version of manuscript. In the future, we will conduct more additional downstream experiments to validate the generated fMRI signals. We will also include the corresponding tasks and analyses, as suggested by the reference you provided, in the appendix.
> > >
> > > Thank you again for your constructive suggestions and valuable reference, which will undoubtedly enrich our future work.

---

### Official Review · Reviewer_wJoL · 2024-11-04

**Soundness:** 3
**Presentation:** 3
**Contribution:** 3
**Rating:** 8
**Confidence:** 3

**Summary:**

This paper proposes a novel approach for generating realistic fMRI data using diffusion models, specifically designed to account for regional interactions and spectral features of the brain. The model captures regional connectivity in a hierarchical structure, where fine-scale signals are conditioned on larger-scale signals. To learn spectral features, it incorporates loss functions that capture fractal characteristics. Results demonstrate improved performance over existing time-series forecasting and diffusion-based models. Additionally, ablation studies validate the effectiveness of each model component.

**Strengths:**

1. The proposed method, which captures connectivity and spectral features, is a novel approach.
2. The method is rigorously validated using multiple benchmarks and ablation studies.

**Weaknesses:**

1. The paper has limited reproducibility due to missing details about data preparation and experimental setup. Additional information is needed on the dataset used, including whether it involved resting-state or task-based fMRI, whether subjects were healthy or under specific conditions, and the rationale for selecting regions of interest (ROI), which were reduced from 268 to 82. Clarification on data split (e.g., train/test division, sample counts) is also required. If the codebase will not be provided, the paper should include a detailed description of the network architecture (such as layer specifications and activation functions) and the training setup for benchmark methods in an appendix.
2. The practical implications, particularly the clinical applications of the proposed method, are somewhat unclear and could benefit from further exploration and discussion.

**Questions:**

1. What is the sampling frequency for the fMRI?
2. It would be beneficial to analyze the reconstructed signal to determine if the observed patterns align with expectations. Calculating the spectrum and fractal characteristics would provide an important validation of the model’s effectiveness. Additionally, some neurological features can also be checked. For instance, if the data is from resting-state fMRI, does it reveal the default mode network?
3. Line 465 typo 69->96

---

> ### Author Response · Authors · 2024-11-20
> **Response to Reviewer wJoL (Part 1)**
>
> Thank you very much to your careful review and valuable comments, which have been extremely helpful to us. Below, we address each of your questions individually and aim to clarify any concerns you may have.
>
> > Question1: What is the sampling frequency for the fMRI?
>
> For fMRI data in HCP dataset, the repetition time (TR) is 720 $\text{ms}$ and the echo time (TE) is 33.1 $\text{ms}$. The voxel size is 2$\text{mm}^3$.
>
> > Question2: It would be beneficial to analyze the reconstructed signal to determine if the observed patterns align with expectations. Calculating the spectrum and fractal characteristics would provide an important validation of the model’s effectiveness.
>
> To evaluate whether generated fMRI signals have expected fractal characteristics, we computed the $q$-order singularity exponent $hq$ and corresponding dimension $Dq$ for both the generated signals and the real fMRI signals. Additionally, we compared our proposed method with other state-of-the-art prediction methods (MSGNet [1], U-Mixer [2]) in terms of fractal characteristics. The results, including those for real fMRI data, are plotted as functions of the order $q$ in Figure 1 in anonymous link [https://anonymous.4open.science/r/iclr_6851/](https://anonymous.4open.science/r/iclr_6851/). The results demonstrate that the fMRI data generated by our proposed method exhibit multifractal characteristics that are closer to those of real fMRI data. This indicates that our model effectively captures the complex hierarchical dynamics and multifractal properties inherent in real neural signals.
>
> The results are averaged over all test subjects and demonstrate that the multifractal characteristics of the fMRI data generated by our method is closer to that of the real fMRI data compared to those generated by other methods. This analysis confirms that our method effectively maintains the multifractal properties of the original brain signals.
>
> > Question2: Additionally, some neurological features can also be checked. For instance, if the data is from resting-state fMRI, does it reveal the default mode network?
>
> To evaluate whether the fMRI signals generated by our model are consistent with real biological fMRI signals in terms of other neurological features, we employ three quantitative measures, following [3].
>
> First, we compute the Pearson Correlation Coefficients (PCCs) between the time series of resting-state fMRI signals from our generative model and those from the real brain. Second, we assess the similarity of the region-wise Functional Connectivity (FC) matrices derived from the generated and biological fMRI signals. This assessment involves calculating the PCCs between two FC matrices and computing the Frobenius norm (F-norm) of the difference between them. The average PCC in the fMRI signals over all ROIs is **0.981**, the PCC between the FC matrices of our generative model and those from the real brain is **0.998**, and the F-norm distance between these FC matrices is **0.586**.
>
> This validates the physiological alignment between the generated fMRI signals and the biological brain data, demonstrating that the generated fMRI signals hold potential for broader applications in various brain analysis tasks.
>
> > Question3: Line 465 typo 69->96
>
> We sincerely apologize for the typo error and greatly appreciate you pointing it out. We will correct it in the paper.
>
>
> [1] Cai, Wanlin, et al. "Msgnet: Learning multi-scale inter-series correlations for multivariate time series forecasting." Proceedings of the AAAI Conference on Artificial Intelligence. Vol. 38. No. 10. 2024.
>
> [2] Ma, Xiang, et al. "U-Mixer: An Unet-Mixer Architecture with Stationarity Correction for Time Series Forecasting." Proceedings of the AAAI Conference on Artificial Intelligence. Vol. 38. No. 13. 2024.
>
> [3] Lu, Wenlian, et al. "Imitating and exploring the human brain's resting and task-performing states via brain computing: scaling and architecture." National Science Review 11.5 (2024): nwae080.

---

> ### Author Response · Authors · 2024-11-20
> **Response to Reviewer wJoL (Part 2)**
>
> > Weakness1: The paper has limited reproducibility due to missing details about data preparation and experimental setup. Additional information is needed on the dataset used, including whether it involved resting-state or task-based fMRI, whether subjects were healthy or under specific conditions, and the rationale for selecting regions of interest (ROI), which were reduced from 268 to 82.
>
> Thank you for pointing out this important issue, and we apologize for the confusion caused by the lack of detailed descriptions in the paper. The data we used in experiments is resting-state fMRI from HCP[4] and subjects are healthy.
>
> In our experiments, we used the Automated Anatomical Labeling (AAL) atlas[5] for brain parcellation on the HCP resting-state fMRI data. The AAL atlas is based on macroscopic anatomical parcellation of the brain and is widely used in neuroscience research. The AAL atlas has multiple versions[5][6], each differing in the number of brain regions. In our experimental setup, we use the version of AAL comprising 116 ROIs. Considering that fMRI analyses commonly focus on brain regions, we selected 82 ROIs corresponding to brain regions for our analysis, excluding the remaining ROIs, which belong to cerebellar and other regions. The number 268 mentioned in our manuscript refers to an intermediate step in our data processing pipeline. We apologize for any confusion this may have caused. We will revise this in the paper to address this point.
>
> For more details on the HCP resting-state fMRI data, you can refer to [4]. To facilitate your understanding, we add additional descriptive details as follows:
>
> The dataset we used in our paper is the Human Connectome Project (HCP)[4], which can be applied to download and use from website [https://db.humanconnectome.org/](https://db.humanconnectome.org/). We followed selection criteria from previous studies to ensure the quality and consistency of the dataset. Specifically, we included right-handed subjects who had completed four runs of resting-state fMRI (rs-fMRI) across two sessions (REST1/REST2), each with two phase-encoding directions (LR/RL), and exhibited minimal movement (mean square displacement < 0.1 mm frame-to-frame motion).
>
> The HCP fMRI data were acquired using a multiband factor of 8, providing a spatial resolution of 2 mm isotropic voxels and a repetition time (TR) of 0.7 s. Each subject underwent two 15-minute resting-state scans, with eyes open and fixating on a crosshair, acquired in opposing phase-encoding directions (LR and RL). This resulted in a total of one hour of resting-state data collected over two visits.
>
> The participants in the HCP dataset are classified as healthy individuals, and detailed information about their characteristics can also be found in [4]. Below, we provide a summary of these details:
>
> The primary participants were recruited from Missouri families with twins, reflecting U.S. ethnic and racial diversity based on the 2000 census. The term "healthy" was defined broadly to include individuals from diverse behavioral, ethnic, and socioeconomic backgrounds. However, certain exclusions were applied to ensure imaging quality and consistency. These exclusions included individuals with severe neurodevelopmental disorders (e.g., autism), neuropsychiatric conditions (e.g., schizophrenia or major depression), neurological disorders (e.g., Parkinson's disease), diabetes, or high blood pressure. Additionally, twins born before 34 weeks of gestation and non-twins born before 37 weeks were excluded from the dataset. Participants with mild behavioral issues, such as smoking, being overweight, or recreational drug use, were included to capture a representative sample of the population.
>
> > Weakness1: Clarification on data split (e.g., train/test division, sample counts) is also required.
>
> From the original HCP sample of 900 subjects, we selected those with valid and complete resting-state fMRI measurements, resulting in a final sample size of 870 subjects. The dataset was then split into 696 subjects for training and 174 subjects for testing.
>
> [4] Van Essen, David C., et al. "The Human Connectome Project: a data acquisition perspective." Neuroimage 62.4 (2012): 2222-2231.
>
> [5] Rolls, Edmund T., Marc Joliot, and Nathalie Tzourio-Mazoyer. "Implementation of a new parcellation of the orbitofrontal cortex in the automated anatomical labeling atlas." Neuroimage 122 (2015): 1-5.
>
> [6] Rolls, Edmund T., et al. "Automated anatomical labelling atlas 3." Neuroimage 206 (2020): 116189.

---

> ### Author Response · Authors · 2024-11-20
> **Response to Reviewer wJoL (Part 3)**
>
> > Weakness1: If the codebase will not be provided, the paper should include a detailed description of the network architecture (such as layer specifications and activation functions) and the training setup for benchmark methods in an appendix.
>
> We will provide the codebase, including the network architecture and training setup for the benchmark methods. Here, we also briefly describe the specific design of the network as presented in our paper.
>
> The denoising network in diffusion model we used in PDH-Diffusion consists of an input projection using a 1D convolutional layer with circular padding, followed by a diffusion time embedding module and a condition upsampling module to align signals. The core is a stack of residual layers with dilated convolutions, Leaky ReLU activations (slope$=$0.4), and skip connections. The outputs from skip connections are aggregated and passed through a final projection layer (a 1D convolutional layer) to reconstruct the denoised signal.
>
> > Weakness2: The practical implications, particularly the clinical applications of the proposed method, are somewhat unclear and could benefit from further exploration.
>
> Thank you for your valuable feedback. To address this concern, we have added three additional experiments.
>
> **First**, to validate the clinical applicability of our method, we conducted experiments on the ABIDE dataset. Considering that the HCP fMRI data used in our experiments are from healthy individuals, this experiment evaluates the generalizability of our method to fMRI data from individuals with certain neurological disorders, thereby demonstrating its potential for broader clinical applications. **Second**, to assess whether the generated fMRI data can support clinical applications, we evaluated their performance on downstream tasks, specifically cognitive prediction. **Finally**, to validate that the generated data can be reliably used for advanced brain analyses and further enhance their potential clinical utility, we evaluated whether our method retains brain network structures consistent with those observed in real brains through several physiological quantitative metrics.
>
> 1. Applicability to Disease-Related fMRI Data:
>
>    Our method is not only effective for fMRI data from healthy individuals but also demonstrates strong performance on data from individuals with certain diseases, such as those in the ABIDE dataset[7].
>
>     The ABIDE dataset compiles fMRI data from multiple international sites, focusing on brain imaging of individuals with Autism Spectrum Disorder (ASD) and Typically Developing Controls (TC). Data were collected across two phases: ABIDE I (2012) and ABIDE II (2016), with over 2,000 participants combined. The dataset includes resting-state fMRI scans from children, adolescents, and adults ages 7 to 64 years old, ensuring a broad demographic representation.
>
>     In the experiment, the context length is 32 and prediction length is 32. The results, as summarized in the table below, highlight the versatility of our method as a general-purpose fMRI generation approach. Our method's ability to generate high-quality fMRI signals for certain patients further underscores its application potential, reducing acquisition time and benefiting downstream tasks such as disease prediction.
>
>     | Method              | MAE $\downarrow$  | RMSE $\downarrow$ | MAPE $\downarrow$ |
>     |---------------------|---------|---------|--------|
>     | MSGNet[1] | 19.95   | 26.84   | 2.69   |
>     | U-Mixer[2]  | 18.74   | 24.97   | 1.55   |
>     | PDH-Diffusion (Ours)       | **18.15**   | **23.64**   | **1.48**   |
>
>     These findings show that our method achieves comparable performance to that of real fMRI data in downstream tasks. The results provide further insight into the effectiveness of our method in generating high-quality fMRI signals that are applicable to real-world tasks.  This reinforces the practical importance of our method in both clinical and research settings.
>
>     Moreover, to intuitively illustrate the functional connectivity (FC) patterns between real and generated fMRI data for both Autism Spectrum Disorder (ASD) and Typically Developing Controls (TC), we visualized the FC in Figure 2 in anonymous link [https://anonymous.4open.science/r/iclr_6851/](https://anonymous.4open.science/r/iclr_6851/). In this visualization, green lines represent matching connections between the real and predicted FC, while red lines indicate discrepancies. As shown in the figure, our predicted data closely aligns with the real data in terms of FC patterns. Additionally, it is evident that individuals with ASD exhibit sparser FC connections compared to typically developing individuals. This underscores the potential of our generated fMRI data to provide reliable reference information for clinical diagnosis.
>
> [7] Heinsfeld, Anibal Sólon, et al. "Identification of autism spectrum disorder using deep learning and the ABIDE dataset." NeuroImage: Clinical 17 (2018): 16-23.

---

> ### Author Response · Authors · 2024-11-20
> **Response to Reviewer wJoL (Part 4)**
>
> 2. Evaluation on Downstream Tasks:
>
>     We further investigate the differences in downstream task performance when using fMRI signals generated by our proposed method compared to real fMRI data.
>
>     Predicting phenotypic characteristics using fMRI data is vital for inferring cognitive abilities, behavioral patterns, and other complex traits. This endeavor is crucial for deepening our understanding of the relationships between brain activity and behavior. It is especially significant in the study of psychiatric and developmental disorders, where variations in brain connectivity may be associated with cognitive or emotional functioning.
>
>     To validate the effectiveness of our predicted fMRI data in cognition prediction, we conducted experiments using the benchmark PTGB dataset[8]. We use real fMRI data for the training phase. For testing, we compare the results using real fMRI data and our predicted data with a prediction length of 64. Table below illustrates the performance of our method across five cognitive prediction tasks: **ReadEng**, **Flanker**, **PicSeq**, **ProcSpeed**, and **SCPT\_SEN**. Each cognitive task was evaluated using four standard metrics: Mean Absolute Error (MAE), Root Mean Square Error (RMSE), Mean Absolute Percentage Error (MAPE), and Correlation Coefficient (CC).
>
>     Table below shows the performance comparison of the downstream cognition prediction tasks using real and predicted fMRI signals.
>
>     | **Task** | **Metric** | **Real fMRI** | **Generated fMRI** |
>     | :-: | :-: | :-: | :-: |
>     | ReadEng | MAE  | 9.9580 | 10.8037 |
>     |  | RMSE | 14.2650 | 15.2069 |
>     |  | MAPE | 8.817  | 9.534 |
>     |  | CC   | 0.347  | 0.273 |
>     | Flanker | MAE  | 6.1559 | 6.333 |
>     |  | RMSE | 9.2298 | 8.979 |
>     |  | MAPE | 5.522  | 5.678 |
>     |  | CC   | 0.314  | 0.331 |
>     | PicSeq | MAE  | 11.4232 | 12.2050 |
>     |  | RMSE | 14.0081 | 15.0200 |
>     |  | MAPE | 11.071  | 11.803 |
>     |  | CC   | 0.333  | 0.290 |
>     | ProcSpeed | MAE  | 9.3325 | 9.4080 |
>     |  | RMSE | 11.3260 | 11.4040 |
>     |  | MAPE | 8.241  | 8.304 |
>     |  | CC   | 0.112  | 0.132 |
>     | SCPT_SEN | MAE  | 0.0708 | 0.0388 |
>     |  | RMSE | 0.1141 | 0.0720 |
>     |  | MAPE | 8.195  | 4.713 |
>     |  | CC   | 0.116  | 0.179 |
>
>     These findings show that our method achieves comparable performance to that of real fMRI data in downstream tasks. This further supports our motivation: by reducing acquisition time and cost, our approach enables the generation of high-quality fMRI data that retains its utility for critical applications, such as disease prediction and cognitive analysis. This reinforces the practical importance of our method in both clinical and research settings.
>
>
> 3. Physiological Metrics of fMRI for Further Analysis:
>
>     As addressed in our response to Question 2, we have conducted the corresponding experiments to evaluate our method.
>
> We hope these clarifications and additional results can address your concerns and demonstrate the broad applicability and significance of our work.
>
> [8] Yang, Yi, Hejie Cui, and Carl Yang. "Ptgb: Pre-train graph neural networks for brain network analysis." arXiv preprint arXiv:2305.14376 (2023).

---

> > ### Comment · Reviewer_wJoL · 2024-11-23
> >
> > Thank you for the response. With the additional information, I think it is a good paper and a great contribution to the field.

---

> ### Author Response · Authors · 2024-11-27
> **Response to Reviewer wJoL**
>
> We sincerely appreciate your thoughtful review and the valuable assistance you've provided for our work. Thank you for your willingness to increase the score!

---

### Meta-Review · Area_Chair_KbRG · 2024-12-12

**Metareview:**

This submission contributes a data-generation model for fMRI time-series. It generate much interest and discussion, with the reviewers appreciating that the method captures both connectivity and time-wise aspects of the fMRI signal, hierarchical regional interactions and multifractal dynamics, into the diffusion process. The reviewers also brought forward the solid empirical study. It was however noted that the practical application of the work is not clear.

**Additional Comments On Reviewer Discussion:**

There was a thorough discussion with back and forth between authors and reviewers. The discussion led to more thorough validation, included in the final manuscript.

---

### Decision · Program_Chairs · 2025-01-22

Accept (Poster)